# Reciprocal Paracrine Signaling and Dynamic Coordination of Transitional States in the Alveolar Epithelial Type 2 Cells and Associated Alveolar Lipofibroblasts During Homeostasis, Injury and Repair

**DOI:** 10.3390/cells14231869

**Published:** 2025-11-26

**Authors:** Georgios-Dimitrios Panagiotidis, Mengqing Chen, Xiuyue Yang, Manuela Marega, Stefano Rivetti, Xuran Chu, Saverio Bellusci

**Affiliations:** 1Department of Medicine V, Internal Medicine, Infectious Diseases and Infection Control, Universities of Giessen and Marburg Lung Center (UGMLC), German Center for Lung Research (DZL), Justus-Liebig University Giessen (JLU), 35392 Giessen, Germany; georgios.panagiotidis@uni-giessen.de (G.-D.P.);; 2Cardio-Pulmonary Institute (CPI), 35392 Giessen, Germany; 3Institute for Lung Health (ILH), 35392 Giessen, Germany; 4Department of Medicine II, Internal Medicine, Pulmonary and Critical Care, Universities of Giessen and Marburg Lung Center (UGMLC), German Center for Lung Research (DZL), Justus-Liebig University Giessen, 35392 Giessen, Germany; 5Department of Respiratory and Critical Care Medicine, Affiliated Hospital of Southwest Medical University, Luzhou 646000, China; chenmq@swmu.edu.cn; 6School of Pharmaceutical Science, Wenzhou Medical University, Wenzhou 325035, China; xiuyue67@gmail.com (X.Y.); chuxuran@gmail.com (X.C.); 7Oujiang Laboratory, Zhejiang Lab for Regenerative Medicine, Vision and Brain Health, Wenzhou 325035, China; 8Department of Pulmonary and Critical Care Medicine, The Quzhou Affiliated Hospital of Wenzhou Medical University, Quzhou People’s Hospital, Quzhou 324000, China; 9Laboratory of Extracellular Matrix and Regeneration, Justus-Liebig University Giessen (JLU), Cardio-Pulmonary Institute (CPI), Member of the German Center for Lung Research (DZL) and Institute for Lung Health (ILH), 35392 Giessen, Germany

**Keywords:** alveolar regeneration, AT2/AT1 transitional cells, lipofibroblast–myofibroblast (LIF–MYF) switch, AREG–EGFR signaling, fibrosis resolution, WI-38 fibroblast model

## Abstract

**Highlights:**

**What are the main findings?**

**What are the implications of the main findings?**

**Abstract:**

Single-cell RNA-sequencing has transformed our understanding of alveolar epithelial type 2 (AT2) cells and alveolar lipofibroblasts (LIFs) during lung injury and repair. Both cell types undergo dynamic transitions through intermediate states that determine whether the lung proceeds toward regeneration or fibrosis. Emerging evidence highlights reciprocal paracrine signaling between AT2/AT1 transitional cells and LIF-derived myofibroblasts (aMYFs) as a key regulatory axis. Among these, amphiregulin (AREG)–EGFR signaling functions as a central profibrotic pathway whose inhibition can restore alveolar differentiation and repair. The *human* WI-38 fibroblast model provides a practical platform to study the reversible LIF–MYF switch and screen antifibrotic and pro-regenerative compounds. Candidate therapeutics including metformin, haloperidol and FGF10 show promise in reprogramming fibroblast and epithelial states through metabolic and signaling modulation. Integrating WI-38-based assays, alveolosphere co-cultures, and multi-omics profiling offers a translational framework for identifying interventions that halt fibrosis and actively induce lung regeneration. This review highlights a unifying framework in which epithelial and mesenchymal plasticity converge to define repair outcomes and identifies actionable targets for promoting alveolar regeneration in chronic lung disease.

## 1. Introduction

The diversity of topics covered in this review, including AT2 heterogeneity, fibroblast activation states, and key signaling pathways, reflects the complexity of alveolar regeneration rather than a shift in focus. Each of these components contributes to a unified framework in which reciprocal epithelial–mesenchymal communication determines whether lung injury resolves through regeneration or sustains toward fibrosis. By integrating molecular, cellular, and pharmacological perspectives, we aim to delineate how convergent cellular and signaling axes coordinate epithelial and mesenchymal fate that define the outcome of lung damage versus repair.

### 1.1. Progressive AT2 to AT1 Differentiation Involves the Formation of Intermediate AT2/AT1 States

Alveoli are the alpha and omega unit for gas exchange [1]. Alveolar epithelial type 2 (AT2)-to-alveolar epithelial type 1 (AT1) differentiation during homeostasis is highly important for gas exchange and for the progression of pulmonary diseases, particularly fibrosis, together with the LIF–iLIF–aMYF (lipofibroblast–inflammatory intermediate lipofibroblast–activated myofibroblast) axis [2,3]. AT2s are the prime source of facultative stem cells during lung regeneration/repair as progenitors for AT1s [1]. Many factors can lead to alveolar epithelial injury [4]. Following injury, AT2s proliferate and transiently dedifferentiate through an intermediate AT2-to-AT1 transitional state, as confirmed by lineage tracing in bleomycin-injured *mice* and single-cell RNA-sequencing of *human* IPF lungs [5,6]. These cells eventually differentiate into mature AT1 cells (Figure 1) [7,8].

The studies summarized in this review are based on several well-established experimental models of lung injury that illustrate the dynamics of epithelial and mesenchymal cell interactions during homeostasis, injury, and repair. Among these, the bleomycin (BLM)-induced lung injury model is the most extensively characterized and serves as a reference for delineating the sequential cellular events that drive alveolar damage and resolution. Following intratracheal or intranasal administration of BLM, the acute inflammatory phase occurs within days 3–7, characterized by epithelial injury, immune cell infiltration, and activation of fibroblasts. The fibrotic phase typically peaks between days 14–21, marked by myofibroblast accumulation, extracellular matrix deposition, and the emergence of transitional epithelial states such as damage-associated transient progenitors (DATPs) and pre-alveolar type I transitional cells (PATS). The resolution phase begins around day 21 and extends to approximately days 28–35, during which inflammatory activity declines, fibroblasts revert toward a lipogenic phenotype, and alveolar type II (AT2) cells progressively differentiate into mature alveolar type I (AT1) cells to restore epithelial integrity. While the bleomycin model remains the most widely used and best characterized system for investigating lung fibrosis, additional experimental approaches provide important complementary insights into progressive and chronic disease mechanisms. Silica and radiation exposure models induce persistent and spatially restricted fibrotic lesions that more closely resemble the irreversible scarring observed in idiopathic pulmonary fibrosis (IPF) [9,10]. Viral infection models, including influenza and SARS-CoV-2, have further revealed epithelial vulnerability, immune dysregulation, and prolonged AT2/AT1 transitional states that underlie post-viral fibrosis [5,6,11]. Moreover, genetic and toxin-induced models, such as surfactant protein C deficiency, telomerase mutations, and elastase- or pollutant-induced injury, demonstrate how epithelial senescence and oxidative stress predispose the lung to chronic fibrotic remodeling [5,6,11,12]. In addition to the BLM model, complementary systems such as elastase-induced emphysematous injury, pneumonectomy (PNX)-driven compensatory regeneration, and viral infection models (including influenza and SARS-CoV-2) have been instrumental in revealing conserved mechanisms of AT2 plasticity, epithelial–mesenchymal crosstalk, and fibroblast reprogramming. Collectively, these models provide a temporal and mechanistic framework for interpreting the cellular and molecular processes underlying alveolar regeneration and fibrosis resolution discussed throughout this review.

### 1.2. From Injury to Identity: Many Names, One Character

In recent years, several studies have described a transitional population of alveolar epithelial cells that emerge during lung injury and repair, though the terminology used varies across research groups. These cells, derived from alveolar type 2 (AT2) cells, exhibit characteristics of both AT2 and AT1 lineages but do not fully resemble either, occupying an intermediate state. Some researchers have termed them “pre-alveolar type-1 transitional cells” (PATS), while others refer to them as “damage-associated transient progenitors” (DATPs), “alveolar differentiation intermediates” (ADIs), or more recently, AT0s. Notably, these transitional cells are not merely driven by injury. They could represent a latent regenerative reservoir accessible therapeutically with the appropriate pharmacological interventions (Table 1).

Using lineage tracing and single-cell RNA-sequencing (scRNA-seq) data, researchers confirmed the presence of transitional cells originating from AT2s and identified distinct molecular signatures associated with this intermediate state. While the majority of these cells express keratin 8 (*Krt8*), subsets also exhibit the expression of claudin 4 (*Cldn4*), keratin 19 (*Krt19*), connective tissue growth factor (*Ctgf*), and stratifin (*Sfn*) [3,5,13,14,15,16]. In the study by [17], the authors uncovered a novel function of the PCLAF (PCNA Clamp Associated Factor; also known as KIAA0101 or PAF)–DREAM (the dimerization partner, retinoblastoma (RB)-like, E2F, and multi-vulval class B) signaling axis in modulating the plasticity of alveolar epithelial cells during the shift from AT2 to AT1 phenotypes. The DREAM complex plays a central role in controlling both the cell cycle dynamics and maintenance of cellular quiescence by fine-tuning the transcriptional activity of specific gene networks. In the study by [17], primed alveolar progenitor cells (PAPCs) were characterized transcriptionally as proliferative intermediate cells, and lineage tracing via *Sftpc^CreERT2^* and *Rosa26^Sun1-GFP^* demonstrated that AT2-derived progenitors label into PAPCs and eventually into AT1 cells. Using this system, *Pclaf* deletion in the AT2-lineage markedly reduced the emergence of *Gfp^+^ Hopx^+^* AT1 cells, confirming that *Pclaf* KO in PAPCs impairs their maturation into AT1 cells. However, the authors demonstrated that activation of Tgf-β signaling is sufficient to restore AT1 cell differentiation even in the absence of *Pclaf* [17]. *Pclaf* KO was also found to be important on immune cell and fibroblast homeostasis, important factors in the progression of fibrosis [2].

**Figure 1 cells-14-01869-f001:**
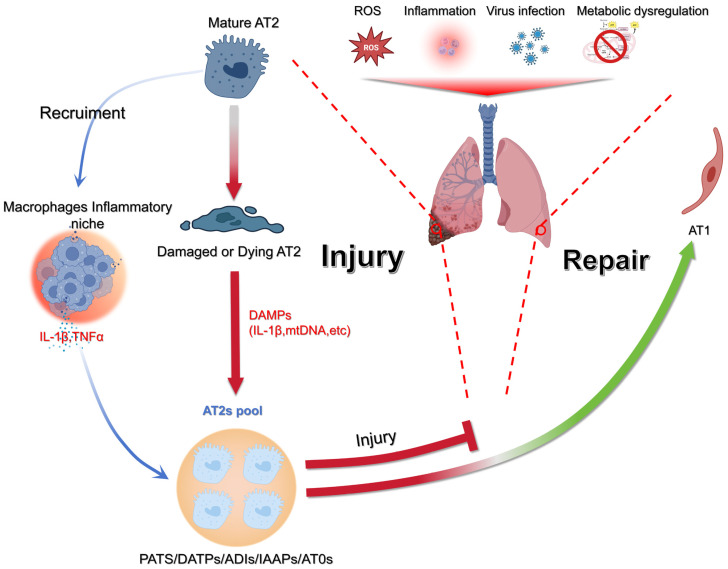
Model of epithelial injury and regeneration in the lung. Environmental and cellular stressors such as ROS, infection, inflammation, or metabolic imbalance can injure AT2s. Once damaged, AT2s release DAMPs (e.g., IL-1β, mtDNA), which stimulate macrophages and create an inflammatory niche enriched in cytokines, including IL-1β and TNFα. This inflammatory response both amplifies tissue injury and signals to the remaining AT2 pool. Surviving AT2s expand and generate transitional progenitor states (PATS/DATPs/ADIs/AT0s), which ultimately affect repair by either replenishing the alveolar epithelium or failing to do so. Successful resolution leads to restoration of alveolar type 1 (AT1) cells, while failure to repair results in sustained injury. Figure created with BioRender.com.

AT2s have been described to be heterogeneous. Among them, captured using the *Sftpc^CreERT2/+^: tdTom^flox/+^ mice*, at least four different subpopulations have been described during homeostasis: the Axin2+ AT2s (aka AEPs or alveolar epithelial progenitors), the Sca1+AT2s, the immature Pd-l1+ AT2s (aka IAAPs or injury-activated alveolar progenitors) and Pd-l1− AT2s [18].

It is difficult at this stage to define which AT2 subpopulation is preferentially engaged in differentiating into the AT1 lineage. The original proposition was that all AT2s can display such capability [1]. However, a major limitation is that the lineage tracing studies using the *Sftpc^CreERT2^* driver line did not discriminate between these four subpopulations. Further experiments using intersectional Dre/Cre-based genetics will be needed to target each of these subpopulations. In addition, in vitro evidence suggests that AEPs display a higher capability to form alveolospheres in vitro compared with the mature AT2 populations [19]. Interestingly, after bleomycin injury, it was described that the mature AT2s underwent massive cell death while the more immature IAAPs were amplified [20]. This suggests that the IAAPs may also represent an important progenitor population to replenish the missing AT2s. Whether the AT2-derived IAAPs can differentiate into AT1 cells is still unknown.

As the AT2 cells differentiate, they begin to acquire AT1 characteristics. An AT2/AT1 transitional state called pre-alveolar type-1 transitional cell state (PATS) [3] and damage-associated transient progenitors (DATPs) are characterized by the expression of *Cldn4*, *Krt8*, N-Myc Downstream Regulated 1 (*Ndrg1*), and Small Proline Rich Protein 1A (*Sprr1a*) [15]. Another group described these intermediate cells as *Krt8+* ADI, exhibiting elevated expression of genes associated with epithelial–mesenchymal transition (EMT), cellular senescence, and key signaling pathways including *p53*, *Myc*, *Tnfα* via *Nf-κb*, and oxidative phosphorylation [5]. More recently, a distinct transitional population in the *human* lung termed alveolar type 0 (AT0) cells has been described. These cells show a hybrid molecular profile, co-expressing AT2 markers such as *SFTPC*, AT1 markers like *AGER*, and secretory cell genes including *SCGB3A2* and *SOX2*, placing them at an intermediate position between AT2, AT1, and bronchiolar secretory cells [21]. Functionally, AT0 cells are thought to originate from AT2 cells and demonstrate bi-potency, with the capacity to give rise either to gas-exchanging AT1 cells or to secretory epithelial cells of the distal airways, thus contributing to progenitor pools at the alveolar–airway interface [22]. Respiratory bronchioles in the *human* lung bridge the terminal bronchioles with the alveolar ducts. Respiratory bronchioles contain respiratory airway secretory (RAS) cells, which have been proposed to be essential for airway repair and regeneration. RAS cells act as progenitors for AT2s [23]. These cells then progressively differentiate into early and late AT1 cells, initially regulated by IL-1β signaling and mainly mediated by HIF1a [15].

The early AT1s express the homeodomain only protein homeobox (*Hopx*) but are negative for insulin-like growth factor-binding protein 2 (*Igfbp2*), while the late AT1 cells are positive for both *Hopx* and *Igfbp2* [24]. Lineage tracing of the early and late AT1s using the corresponding driver lines indicate that in the context of bleomycin injury, only the early but not the late AT1s were capable of differentiating into AT2 cells, refining the original concept that mature AT1s could differentiate into AT2s following injury [15].

From a mechanistic point of view, Interleukin 1 β (IL-1β) has been shown to prevent the differentiation of the AT2/AT1 cells toward the AT1 phenotype, leading to the accumulation of these intermediate cells [25]. This observation suggests that inflammatory processes are likely crucial for maintaining this intermediate state. This observation is counter-intuitive since anti-inflammatory steroid-based therapies have been shown to be ineffective in treating lung fibrosis in *humans* [26,27]. Interestingly, AT2 to AT2/AT1 differentiation is associated with the downregulation of signaling mediated by Fibroblast growth factor receptor 2 b [28], which was previously described to be critical for the survival of the AT2 cells [29]. In addition, the FGFR2b ligand FGF10 displays therapeutic activity in the context of bleomycin-induced fibrosis [29,30,31,32]. Therefore, treatments aiming at preserving or amplifying the AT2 population for patients at risk of developing fibrosis may represent a viable option.

**Table 1 cells-14-01869-t001:** Representative table of names, markers and references of published alveolar epithelial type 2/1 cells.

Name	Markers	Reference
Pre-alveolar type-1 transitional cells (PATS)	Krt8, Krt18, Krt19, Cldn4, Ctgf, Sfn	[3]
Damage-associated transient progenitors (DATPs)	Cldn4, Krt8, Ndrg1, Sprr1a	[15]
Alveolar differentiation intermediates (ADIs)	Krt8+, p53, Myc, Tnfα	[5]
AT0	SFTPC, AGER, SCGB3A2, SOX2	[21]
RAS	SCGB3A2, SCGB1A1	[23]

### 1.3. The Chicken or the Egg Paradigm: In the End, It Is the Epithelium

A first line of thought proposed that dysregulated proliferation and differentiation of the mesenchymal cells, leading to the deposition of high levels of extracellular matrix proteins, including collagens, was the primary cause of fibrosis. Such dysregulation led to epithelial defects, including the loss of AT2s and bronchiolization. However, based on the findings of Gunther and colleagues in [33], chronic injury to the epithelium was proposed as the causative factor for fibrosis formation [33]. This chicken or the egg paradigm regarding the origin of the fibrosis phenotype has been debated for many years. However, in 2020, a final conclusion was suggested through the inactivation of *Cdc42* in AT2 cells using the *Sftpc^CreERT2^* driver line [34]. *Cdc42* is a gene encoding for a Rho kinase involved in numerous cellular processes, including the regulation of actin cytoskeleton dynamics, establishment of cell polarity, shaping of cell morphology, and control of migration. It also contributes to endocytosis, exocytosis, cell cycle progression, and proliferation in a wide range of cell types. Importantly, *Cdc42* has been implicated in the development of various age-related diseases, such as neurodegenerative and cardiovascular conditions, type 2 diabetes, and disorders affecting joints and bones [35]. Loss of *Cdc42* leads to failed AT2-to-AT1 differentiation, accumulation of transitional AT2/AT1 intermediates, and impaired alveolar regeneration. In this model, sustained mechanical tension within the epithelial compartment activates a TGF-β–dependent stress program in AT2 cells, driving progressive fibrotic remodeling [34]. Although this study showed increased p-SMAD2 signaling in adjacent PDGFRβ^+^ stromal cells, the fibroblast subsets involved were not investigated. In contrast, fibroblast lineage-tracing studies demonstrate that TGF-β signaling can induce a reversible transition between LIFs and aMYFs, identifying fibroblast plasticity as a central component of fibrotic progression and resolution [36]. Thus, these findings suggest that epithelial mechanical dysfunction and fibroblast phenotypic transitions act to drive fibrosis, but their precise interdependence remains to be defined.

### 1.4. Reversible and Progressive LIF to MYF Differentiation Switch During Fibrosis Formation and Resolution

Lineage tracing of the activated myofibroblast in young and old *Tg(Acta2-CreERT2); tdTom^flox^ mice* was recently carried out in the context of fibrosis formation and resolution [37,38]. These in vivo lineage-tracing data establish that fibroblast plasticity and partial reversion occur within the injured lung, complementing transcriptomic evidence from *human* fibrotic tissue. The published results demonstrated that during fibrosis formation, there is a transition from alveolar LIF to aMYF. These aMYFs exhibit a fibrotic signature and progressively differentiate from *Cthrc1*^low^ LIF^high^ to *Cthrc1*^high^ cells. During fibrosis resolution, the reverse transition was observed, wherein *Cthrc1*^high^ cells reverted to *Cthrc1*^low^ LIF^high^ and eventually to Acta2-negative LIF^high^ fibroblasts to *Cthrc1^Low^* [37]. More recently, using the *Scube2^CreERT2^* driver line to lineage trace the alveolar fibroblast, a novel intermediate population preceding the formation of the activated myofibroblasts was described. This population, called inflammatory alveolar fibroblasts, is negative for *Acta2* expression and expresses chemokines such as (*Cxcl12)*, serum amyloid A3 (*Saa3*), and lipocalin 2 (*Lcn2*). Their induction is driven by pro-inflammatory cytokines, including interleukin-1β (IL-1β) and tumor necrosis factor (TNF). Inflammatory alveolar fibroblasts were proposed to form from alveolar fibroblasts upon IL17R activation and constitute a transient/intermediate population, which following exposure to TGF-β, subsequently differentiate into *CTHRC1^+^* myofibroblasts (Figure 2) [39].

### 1.5. Use of WI-38 Cells to Investigate LIF to MYF Reversible Switch

The LIF to MYF reversible switch is well described by our group [36,40,41]. This reversibility has been demonstrated in vitro using *human* WI-38 fibroblasts, providing a mechanistic framework that aligns with in vivo fibroblast lineage-tracing studies showing comparable transitions in *murine* models [37,38]. During fibrogenesis, lipofibroblasts can serve as progenitors for activated myofibroblasts. Conversely, in the resolution phase, myofibroblasts have the potential to revert to a lipofibroblast-like state. Pharmacological induction of a lipogenic program has been shown to antagonize TGF-β-driven myogenic differentiation, underscoring a potential therapeutic strategy for idiopathic pulmonary fibrosis (IPF) [36,37,38,39,40,41]. Importantly, the study by [40] emphasized the value of developing cell-based models to investigate the effects of candidate pharmacological interventions on the reversible transition between LIFs and aMYFs, with the ultimate goal of promoting fibrosis resolution.

To summarize, findings from the WI-38 model highlight its value not only as a *human*-derived mesenchymal platform but also as a conceptual bridge connecting cellular metabolism, fibroblast plasticity, and epithelial–mesenchymal interplay. Unlike isolated fibroblast or epithelial cultures, the WI-38 system enables the controlled dissection of paracrine signaling loops that influence both lipogenic and myogenic fibroblast phenotypes. Its responsiveness to pharmacologic agents pinpoints how metabolic reprogramming and lipid homeostasis can be leveraged to reverse fibrotic activation. This model also opens up new perspectives for studying fibroblast dynamics across different organs [42] and could, in the future, be expanded to include additional cell types, such as immune cells, to more accurately recapitulate the complex multicellular interactions that occur under pathophysiological conditions. Therefore, beyond reproducing established findings, the WI-38 paradigm provides a complex, interesting, experimental framework to mechanistically link fibroblast phenotype transitions with epithelial regenerative capacity, advancing a testable model for drug discovery and translational validation.

### 1.6. WI-38 as a Screen for FDA-Approved Drugs

WI-38 cells, a well-characterized *human* embryonic lung fibroblast line, offer a practical and accessible system for earlystage drug screening in the context of pulmonary fibrosis. These cells are responsive to lipogenic stimuli such as metformin, inflammatory stimuli, such as IL-17, and fibrotic stimuli, including TGF-β, making them suitable for modeling the transition between lipofibroblasts, intermediate inflammatory fibroblasts, and activated myofibroblasts. This feature is particularly useful for testing whether FDA-approved compounds can reverse or prevent myofibroblast activation. By combining in vitro assays with transcriptomic and proteomic analysis, WI-38 cells can serve as a first-pass platform to identify promising pharmacological candidates before moving into more complex models, such as lung organoids or in vivo systems. Their ease of culture and reproducibility make them a valuable tool in the search for drugs that could promote fibrosis resolution through lipogenic reprogramming. One interesting FDA-approved drug that can be repurposed for the treatment of fibrosis is haloperidol. Haloperidol, a well-established antipsychotic medication approved by the FDA in the 1960s, is primarily used to treat conditions such as schizophrenia, acute psychosis, and severe agitation. Emerging research suggests that this drug may also have applications beyond the field of psychiatry. A recent high-throughput screening study identified haloperidol as a strong candidate for targeting fibrotic pathways. In fibroblast cultures, it was shown to reduce the expression of *α-SMA*, a key marker of myofibroblast activation. This effect appears to be mediated through its interaction with the sigma-1 receptor, which influences intracellular calcium levels and triggers a mild form of endoplasmic reticulum stress. These intracellular changes ultimately disrupt Notch1 signaling, leading to reduced myogenic differentiation. What makes these findings particularly compelling is that haloperidol also showed antifibrotic effects in animal models of lung, heart, and tumor-associated fibrosis [43]. Given its clinical use and well-documented safety profile, haloperidol may offer a valuable starting point for repurposing efforts aimed at resolving fibrosis. More recently, dopamine receptor agonists, including fenoldopam (D1R-selective), have been found to mitigate myofibroblast differentiation and TGF-β/Smad2 signaling in bleomycin-induced pulmonary fibrosis models, supporting a novel neuromodulatory mechanism of fibrosis control [44]. These findings point to an unexpected but exciting possibility that antipsychotic medications, long used in mental health, might also help tackle fibrosis. Drugs like haloperidol, which have well-known safety profiles, appear to interfere with the same cellular pathways that drive fibrotic remodeling. While much remains to be tested, especially in the clinical setting, this kind of repurposing, together with the use of WI-38-based platforms, could offer a faster, more cost-effective route to developing new antifibrotic therapies.

The WI-38 fibroblast model provides a reproducible and tractable system to study fibroblast plasticity and the LIF–MYF transition, as well as to evaluate pharmacological agents that modulate mesenchymal activation. However, this model alone does not capture the epithelial injury component that initiates and sustains fibrotic remodeling. To address this limitation, the WI-38 system can be integrated with epithelial injury modules using alveolosphere co-cultures or conditioned media derived from injured epithelial cells. These injury paradigms may include exposure to inflammatory cytokines such as IL-1β or TNFα, profibrotic mediators like TGF-β1, proteolytic stress induced by elastase, oxidative injury, or cigarette smoke extract, and mechanical stretch to mimic ventilator-associated stress. Such an integrated approach allows for the simultaneous assessment of epithelial damage, differentiation blockade, and fibroblast activation within a controlled experimental context. The combined WI-38–alveolosphere platform thereby provides a physiologically relevant framework to study reciprocal epithelial–mesenchymal interactions during injury and repair and evaluate therapeutic agents that restore homeostatic signaling and promote alveolar regeneration.

### 1.7. Investigating the Impact of Drugs to Induce LIFs

To develop WI-38 cells into a robust screening platform for FDA-approved drugs aimed at inducing a lipofibroblast phenotype, several molecular targets show promise. One clear example is metformin, an FDA-approved anti-diabetic drug, which has demonstrated clear lipogenic effects in WI-38 cells [40,41]. One key regulator is peroxisome proliferator activated receptor γ (PPARγ), a nuclear receptor that governs lipid metabolism and adipogenic differentiation. Rosiglitazone, a PPARγ agonist, has consistently inhibited TGF-β-induced myofibroblast differentiation in *human* lung fibroblasts, as demonstrated by marked reductions in *α-SMA* and collagen production [36,45]. This is also supported by in vivo studies that confirmed rosiglitazone’s ability to prevent hyperoxia-induced LIF-to-aMYF transdifferentiation in neonatal rat lungs [46]. Nifuroxazide, an anti-diarrheal drug, suppresses the expression of inflammatory factors such as *IL-6* and *IL-4* along with α-SMA and collagen I expression in both TGF-β-stimulated *human* and bleomycin-injured *mouse* fibroblast models, ameliorating pulmonary fibrosis [47]. Saracatinib (a Src-family kinase inhibitor, initially developed as an anti-metastatic drug for cancer treatment [48] and later repurposed for Alzheimer’s disease [49,50]) is also an interesting target for fibrosis resolution. Transcriptomic analysis of *murine* lung extracts showed that saracatinib reverses several key fibrotic pathways, including those involved in epithelial-to-mesenchymal transition, immune signaling, and extracellular matrix remodeling. Notably, its ability to reduce fibrosis and dampen inflammatory responses was further validated in *human* precision-cut lung slices, underscoring its potential as a therapeutic option for lung fibrosis (Figure 3) [51].

### 1.8. Investigating the Impact of Drugs on AT2s

The differentiation pathways of alveolar epithelial cells remain incompletely understood and are still the subject of active discussion. In the *human* lung, trajectory analyses indicate that AT2 cells can transition through an intermediate AT0 state (*SFTPC+, AGER+, SCGB3A2+,* and *SOX2+*) toward mature AT1 cells. This transitional concept aligns with earlier descriptions of *mouse* AT2/AT1 intermediates such as PATS, ADIs, IAPs, and *Krt8+* transitional cells, which share overlapping features but differ in experimental context and nomenclature.

*Human* AT0 cells, like these other *mice* intermediates, exhibit high transcriptional entropy, consistent with their transient and unstable identity. Pseudotime analyses support a model in which AT2-derived AT0s can give rise to AT1s [21]. In *human* AT2-derived alveolosphere cultures, *EGF* withdrawal, but not *FGF10* depletion, promoted AT0-associated markers (*SCGB3A2*, *SFTPC*, *SFTPB*) and induced multiluminal cystic remodeling, indicating an EGF-dependent AT2-to-AT0 differentiation. In vivo, AT0s arise in distal alveolar regions in response to lung injury and are most abundant in mildly fibrotic areas, positioned alongside normal AT2 cells, consistent with their role as an intermediate state [21]. Their presence has also been identified in *human* IPF tissue and reproduced in alveolosphere and organoid systems, confirming cross-species conservation of this transitional state [21]. Pharmacological studies confirmed EGFR involvement: erlotinib and inhibitors of downstream RAF and MEK pathways induced AT0 features, underscoring EGFR’s regulatory role in AT2-to-AT0 transitions. These insights provide a framework for investigating how pharmacological interventions influence AT2 plasticity [21]. Additionally, nintedanib, being an EGFR inhibitor [52], may also have an AT0-promoting effect on AT2s. TGF-β signaling, a key driver of fibrotic remodeling, can be targeted by ALK5 inhibitors, which block TGF-β-induced epithelial-to-mesenchymal transition (EMT) [53,54]. These inhibitors may also alter AT0 and AT1 phenotypes during lung injury repair.

### 1.9. Investigating AT2/LIF Dynamic Interaction Using Alveolospheres

We have previously reported that we can generate alveolospheres using *human* WI-38 cells recombined with *mouse* AT2 cells. It is worth mentioning that this model may reveal limitations in cross-species paracrine compatibility; however, on the other hand, it provides the advantage of tracing the transcriptomic activity of two different species population-wise. We used AT2s from *Sftpc^GFP^ mice* for better visualization of the differentiation status of alveolar epithelial cells. The differentiation of WI-38 cells towards the MYF or LIF can also be manipulated before assembling them with AT2s. We demonstrated that WI-38 cells differentiated to aMYF display a weaker niche activity for AT2s compared to LIF, with a reduction in both the size and colony-forming efficiency [40].

In addition, the epithelial–mesenchymal interactions occurring in the alveolospheres to drugs can also be assessed once they are formed. Using immunofluorescence (IF) and flow cytometry, it is possible to examine the impact of pro-fibrotic treatment (TGF-β) and anti-fibrotic treatment (TGF-β1 + antifibrotic treatment) on the epithelium and mesenchyme. In addition, using bulk RNA-seq on the whole organoid, bioinformatic analysis based on the fact that epithelial and mesenchymal cells arise from different species, allows us to pinpoint separately transcriptomic changes occurring in the epithelium (*mouse* genome) and mesenchyme (*human* genome) over time. These combined approaches will help to better define the mechanism of action of given drugs.

### 1.10. Devising Therapeutic Intervention to Resume AT2/AT1 Differentiation Towards AT1 or Reverse AT2/AT1 Differentiation Towards AT2

Persistence of ‘’transitional’’, ‘’intermediate’’, and ‘’stalled’’ AT2/AT1 states has been implicated in impaired alveolar regeneration and progressive fibrotic remodeling [34], making the promotion of efficient AT2-to-AT1 maturation an attractive therapeutic strategy. Wnt signaling has also been shown to play a major role in the AT2 to AT1 differentiation axis, with *Axin2* functioning as a molecular switch. Axin2 expression is downregulated as AT2s transdifferentiate into AT1 cells. Sustained Wnt signaling prevents this transdifferentiation, whereas the inhibition of Wnt signaling facilitates it [55]. Thus, Axin2 inhibitors would tilt the balance towards AT1 differentiation, and tankyrase inhibitors, which inhibit the degradation of Axin2, might potentially lead to the dedifferentiation of AT1s back to AT2s. Retinoic acid is another interesting factor. Retinoids have been found to play a crucial role in alveologenesis and restoration following injury [56], with a direct positive effect on the number of AT2 cells [57].

The Yap/Taz axis, a member of the Hippo signaling pathway, is also an interesting target, as it has been found to positively affect alveolar regeneration after *Streptococcus pneumoniae*-mediated [58] inflammation [59]. In this study, *mice* with Yap/Taz-deficient AT2s exhibited a delay in alveolar epithelial repair, whereas in wild-type animals, increased AT2 activity was associated with the nuclear expression of Yap/Taz [59]. Given their ability to promote Yap activation [58], microtubule-disrupting agents such as vinorelbine, vincristine, vinblastine, mebendazole, colchicine, and podofilox may be of interest for supposedly promoting AT2 cell regeneration, potentially by inducing the dedifferentiation of intermediate alveolar epithelial cells back into the AT2 lineage.

### 1.11. Nintedanib Target: AREG/EGFR Signaling Takes Central Stage

Amphiregulin (AREG) is a member of the epidermal growth factor (EGF) family and was first identified over 25 years ago. As described by its name (amphi meaning “on both sides”), it exhibits an anti-proliferative function in certain carcinoma cell lines and a pro-proliferative function in fibroblasts [60]. Current insights suggest that AREG promotes proliferation by binding to the EGF receptor (EGFR) [61], thereby initiating tyrosine kinase signaling cascades, including MAPK, PI3K/AKT, mTOR, [61] and PLC [62,63]. These signaling pathways regulate gene expression and trigger various cellular responses, including proliferation, survival, motility, and angiogenesis [64]. Through the signaling pathways mentioned above, AREG is linked to the promotion of fibrosis through TGF-β [65]. Furthermore, AREG also plays a role in airway inflammation and asthma [66,67].

Interestingly, Ref. [6] demonstrated a link between AREG expression and intermediate alveolar type 2 (AT2) cells during fibrotic progression. This association has been validated in vivo in bleomycin-injured *mouse* lungs and corroborated by *human* single-cell datasets, indicating its physiological relevance [6,11]. Their findings revealed that in fibrotic regions of both *murine* models and lungs from patients with idiopathic pulmonary fibrosis (IPF), AT2 cells exhibited increased expression of intermediate AT2/AT1 markers alongside elevated *Areg* levels [6]. Similar observations were previously reported in a bleomycin-induced lung injury model [5]. In addition, AREG has been found to be essential for the progression of fibrosis, also in the context of fibroblast activation [11], even though it is not required for the differentiation of AT2 cells into the intermediate state. These profibrotic effects are mediated via EGFR signaling, as pharmacological inhibition of EGFR abrogated the actions of AREG [6].

Nintedanib is a tyrosine kinase receptor inhibitor [12] and an anti-fibrotic medication that limits the progression of the disease [68]. In addition to other growth factor receptors, nintedanib also exhibits affinity for EGFR in certain contexts, thereby inhibiting its function [52]. In vitro evidence from cancer research indicates that nintedanib reduces AREG levels by inhibiting EGFR signaling [69]. Thus, the link between them is indirect but mechanistically aligned as nintedanib can attenuate stalled intermediate AT2/AT1 driven fibrosis mediated by AREG, ultimately leading to alveolar regeneration.

In addition to its established profibrotic role, AREG also exerts context-dependent effects that influence both epithelial repair and mesenchymal activation. In chronic or unresolved injury, sustained AREG–EGFR signaling drives fibroblast proliferation, myofibroblast differentiation, and extracellular matrix deposition, thereby perpetuating fibrosis [6,11]. Single-cell transcriptomic analyses have revealed that AREG expression is enriched in transitional AT2/AT1 cells and activated myofibroblasts, suggesting that these cell populations may form a self-reinforcing paracrine loop that sustains pathological remodeling [5,6]. Thus, the amplitude and duration of AREG–EGFR signaling determine whether the outcome favors regeneration or fibrosis. Understanding this temporal regulation will be essential for designing targeted interventions that attenuate profibrotic AREG activity while preserving its beneficial reparative functions [5].

### 1.12. LOX and PCSK9 as Upstream Modulators of the Fibroblast–Epithelium Axis

Although no studies have directly examined LOX or PCSK9 in the context of LIF biology or AT2–LIF paracrine signaling, both pathways influence upstream processes that shape the fibrotic niche. Lysyl oxidase (LOX) increases extracellular matrix crosslinking and stiffness, which is known to enhance fibroblast activation and promote profibrotic mechanical feedback [70,71,72,73]. Proprotein convertase subtilisin/kexin type 9 (PCSK9), through its regulation of lipid homeostasis, oxidized LDL uptake, inflammation, and epithelial stress responses, has been shown to modulate EMT, oxidative injury, and Wnt/β-catenin signaling in alveolar epithelial cells [42,74,75,76,77,78]. While their direct involvement in LIF-to-MYF transitions remains untested, these pathways represent broader matrix–lipid regulatory axes that may indirectly influence epithelial–mesenchymal reciprocity in fibrotic lung remodeling. Their integration into future mechanistic studies could clarify whether they contribute to LIF or transitional AT2 cell dysfunction.

### 1.13. Targeting MicroRNAs to Restore Alveolar Differentiation Dynamics

Small non-coding RNAs, known as microRNAs (miRNAs), have emerged as pivotal regulators of lung epithelial cell fate, primarily by modulating mRNA stability and translation [79,80]. In the context of lung injury and fibrosis, specific miRNAs have been found to influence whether transitional alveolar epithelial cells continue differentiation toward a mature AT1 phenotype or revert to a progenitor-like AT2 state. Depending on the signaling context, these miRNAs can either facilitate tissue regeneration or exacerbate pathological remodeling [81,82,83,84,85,86,87,88,89]. Deciphering how individual miRNAs shape alveolar epithelial plasticity may provide new avenues for therapeutic intervention aimed at promoting regeneration while limiting fibrotic progression.

According to the study by [89], AT2 cells isolated from patients with IPF exhibited impaired differentiation into AT1 cells. However, transfection of these cells with microRNA-200b-3p and microRNA-200c-3p restored their ability to undergo differentiation into AT1 cells without affecting their proliferation [89]. In the context of alveolar morphogenesis, two studies have shown that miR-17-92 and miR-106b-25 [81] and miR-29 [88] are important for the differentiation in developing lungs. The findings of [81] revealed that miR-17-92 and miR-106b-25 in lung epithelial explants disrupted FGF10-induced budding morphogenesis, a phenotype that was rescued by supplementation with miR-17. Reduced miRNA levels decreased E-Cadherin expression and altered its localization while increasing β-catenin activity and upregulating downstream targets (*Bmp4, Fgfr2b*), and identified *Stat3* and *Mapk14* as direct targets of these miRNAs [81]. These findings suggest that these miRNAs play a critical role in embryonic lung development and should be studied in the context of disease and alveolar differentiation. In addition, the study by [88] revealed that miR-29 family expression increases in *mouse* fetal lung epithelial cells during alveolar morphogenesis and in *human* fetal lung (HFL) explants during type II cell differentiation. In cultured HFL epithelial cells, miR-29 was upregulated, while hypoxia suppressed it, with an inverse correlation with TGF-β2 [88]. Adding to that, the study of [84] revealed that silica exposure drives AT2 cells toward an epithelial–mesenchymal transition (EMT)-like state while simultaneously suppressing miR-29b. miR-29b directly dampens TGF-β1 signaling. In vitro, restoring miR-29b levels in RLE-6TN cells not only reversed EMT but also promoted mesenchymal–epithelial transition (MET), a switch accompanied by reduced TGF-β1 activity. These in vitro findings translated to silicotic *mice*, where miR-29b supplementation not only halted fibrosis progression but also improved lung function [84]. Beyond its alveolar morphogenic and anti-fibrotic effects, miR-29b’s modulation of TGF-β1 raises intriguing questions about its potential role in AT2-to-AT1 differentiation, a process where balanced TGF-β signaling is known to be critical [90,91,92]. This dual functionality positions miR-29b as a compelling candidate for targeting both pathological fibrosis and impaired alveolar regeneration. Evidence from [93] demonstrates that members of the miR-29 family are downregulated in AT2s following bleomycin-induced lung injury. This reduction inversely correlates with the elevated expression of extracellular matrix (ECM) components, indicating the presence of fibrosis. The molecular driver of miR-29 suppression was identified as TGF-β1. Conversely, miR-29 overexpression attenuated fibrotic gene expression in AT2s [93]. Given that TGF-β1 signaling is a known inhibitor of AT2-to-AT1 transdifferentiation [90,91,92], the capacity of miR-29 to antagonize this pathway suggests a potential role in promoting alveolar epithelial maturation. However, direct evidence for miR-29′s involvement in AT1 differentiation remains an open question requiring targeted investigation.

### 1.14. Rethinking AT1 Cell Plasticity: A Conceptual Framework

While current literature lacks definitive evidence that mature alveolar type 1 (AT1) cells dedifferentiate into AT2 cells under physiological conditions, the existence of intermediate AT2/AT1 transitional states (e.g., PATS, ADIs, AT0) [3,5,15,18,21] raises an intriguing possibility. These intermediates, which co-express markers of both lineages, could be pharmacologically directed toward either AT2 or AT1 fates. If such intermediates can be pushed back to an AT2 phenotype, could similar reprogramming be achieved starting from fully differentiated AT1s? Philosophically, this question challenges the traditional view of AT1 terminal differentiation and asks whether targeted interventions (e.g., WNT/FGF modulation, YAP/TAZ activation) might unlock latent plasticity in AT1 cells.

In this scenario, it could be worth philosophically reconsidering the axis of the alveolar epithelial cell fate as recent findings give important yet intriguing information. The study of [30] adds complexity to our understanding of alveolar epithelial cell fate. In a bleomycin-induced lung fibrosis model, the authors identified injury-activated alveolar progenitors (IAAPs) as a distinct cell population. At baseline (day 0), IAAPs were scarce, but their numbers rose sharply during disease progression, coinciding with a decline in AT2 cells, likely due to injury-induced cell loss. By day 16, IAAPs reached their peak abundance, while AT2 cells were at their lowest levels. During the subsequent repair phase, this relationship reversed, with IAAP numbers declining as AT2 cells recovered, a trend that continued until day 60 [30]. These findings raise a key question: do AT2 cells primarily regenerate through self-replication following injury, or do IAAPs function as a progenitor reservoir that replenishes the AT2 population?

Supporting this possibility, recent work by [94] examined the molecular status of *KrasG12D*, a constitutively active mutant form of the *Kras* oncogene, in AT1 cells using single-cell RNA-sequencing (scRNA-seq). This analysis revealed a molecular trajectory consistent with direct reversion from an AT1 to an AT2-like phenotype, suggesting that *KrasG12D* activation can unlock plasticity in mature AT1 cells [94].

To follow up on these ideas, in vitro studies that isolate primary AT1 cells and expose them to pro-regenerative signals such as WNT signaling manipulation, FGF10 supplementation, or YAP/TAZ activation might reveal latent plasticity. Thus, while AT1-to-AT2 dedifferentiation remains unconfirmed, the dynamic nature of alveolar epithelial transitions suggests that the boundaries between terminal and plastic states may be more permeable than previously assumed.

### 1.15. Guardians of the Alveolus: FGFR2 and the Fate of AT2 Cells

FGF signaling promotes proliferation and is essential for lung branching morphogenesis and repair following injury, with FGFR2 playing a critical role in epithelial patterning during development [95,96,97,98]. In adult lungs, FGF signaling is required for AT2 cell maintenance, as demonstrated by studies showing that FGFR1, FGFR2, and FGFR3 support AT2 cell survival and proliferation during homeostasis and post-injury repair [19,31,32,99]. Accordingly, FGFR signaling plays a major role in both epithelial morphogenesis and regeneration. Recent evidence using conditional FGFR2 deletion in AT2 cells (via *Sftpc-Cre*) reveals that this pathway is essential for their self-renewal and ability to restore alveoli after injury. When FGFR signaling is disrupted, AT2 proliferation is inhibited, thereby compromising alveolar regeneration, which demonstrates a cell-intrinsic mechanism. Organoid experiments confirm that FGFR activity governs AT2-to-AT1 differentiation, further supporting its role in driving alveolar regeneration. Complementary in vivo conditional-*Fgfr2b* knockout studies have confirmed the essential role of FGFR2 in epithelial survival and alveolar regeneration [29]. While representative images of these organoids are not presented here due to the review format, their morphology have been described [100]. Moreover, in vivo studies using bleomycin injury models have shown that the deletion of *Fgfr2* alone in AT2 cells leads to increased mortality and lung injury, indicating that FGFR2 is the dominant receptor mediating AT2 survival during injury [29]. Taken together, the data underscore the importance of FGFR2 in preserving AT2 cell function and facilitating epithelial recovery in the injured lung.

### 1.16. Functional Diversity of Alveolar Type II Cells: Coordinators of Regeneration and Immune Homeostasis

The interplay between mesenchymal, epithelial, and immune cells in the lung has emerged as a key determinant of tissue homeostasis and pathophysiological responses. Among these, fibroblasts and alveolar type 2 (AT2) epithelial cells occupy a central position, not only due to their physiological roles but also because of their capacity to influence the behavior of immune cells. Fibroblasts, in particular, are capable of modulating local immune activity through the release of cytokines, chemokines, and matrix-bound signals, thereby regulating both immune and AT2 cell function [2]. AT2 cells, in turn, contribute to this network through immunomodulatory signaling and by maintaining epithelial–stromal balance under both physiological and pathological conditions [25,101,102,103,104,105,106,107,108,109,110].

Detailed mechanisms of fibroblast–immune cell interactions have been extensively reviewed in our earlier work [2]. Here, we focused on immune-related functions of AT2 cells. In a co-culture model of THP-1 macrophages and A549 epithelial-like carcinoma cells, unstimulated epithelial cells were sufficient to induce NF-κB, a major regulator of the immune response, in macrophages [111]. Specifically, 29 out of 84 NF-κB-associated genes were upregulated ≥4-fold, including cytokines, chemokines, and immune receptors accompanied by increased IL-6, IL-8, and ICAM-1 expression [103].

AT2s express and respond to Toll-like receptor (TLR) signaling. TLR2, in particular, plays a key role in initiating signaling cascades that shape downstream inflammatory responses [104]. In *mice*, profiling of the alveolar epithelial cells’ secretome revealed 12 cytokines, chemokines, and growth factors including Gm-csf, Mcp-1, Il-6, and Ip-10, present under basal conditions, with others, such as G-csf, M-csf, and Rantes induced only upon LPS stimulation. Alveolar epithelial cells demonstrate the ability to respond to both microbial and endogenous signals, producing a range of immunomodulatory factors that can influence monocytes, macrophages, dendritic cells, and T cells. Notably, AECs secreted T cell-attracting chemokines such as Ip-10 and Rantes following activation [102].

AT2s play a central role in lung repair, but their function is highly sensitive to the inflammatory state of the surrounding microenvironment. IL-1β, elevated during COPD exacerbations, exerts timing-dependent effects on AT2s’ behavior. While acute IL-1β exposure transiently enhances organoid growth via NF-κB signaling, prolonged exposure disrupts AT2 differentiation and repair capacity. This shift is marked by the accumulation of transitional progenitor states and the altered expression of key markers, including upregulation of *Tm4sf1* and suppression of *Lmo7,* indicating injury-induced AT2/AT1s [25]. Beyond their role in maintaining alveolar structure and participating in tissue repair, AT2 cells also serve as pivotal regulators of pulmonary immune responses during infection. In response to vaccinia virus infection, AT2s upregulate IFN-β and downstream interferon-stimulated genes through MDA5 and STING-dependent nucleic acid sensing. This response promotes the recruitment of *Ccr2^+^* inflammatory monocytes, which differentiate into *Lyve1^−^* interstitial macrophages capable of clearing viral particles and limiting viral replication [108]. During SARS-CoV infection, AT2 cells upregulate type I and III interferons (notably IFN-β and IL-29) and secrete a range of chemokines, including CXCL8 (IL-8), CCL5 (RANTES), CXCL10, and CXCL11, which recruit and activate various immune cell populations, such as neutrophils, monocytes, NK cells, and T lymphocytes, thus orchestrating the early innate immune response [110]. In conclusion, these findings highlight the dual identity of AT2s as both regenerative and immunologically active components of the lung microenvironment. Positioned at the intersection of epithelial, mesenchymal, and immune signaling, AT2 cells not only respond to injury and infection but also actively orchestrate immune cell recruitment and function.

IAAPs have also been identified to have immunoregulatory functions [18,29,112]. ATAC-seq analysis upon pneumonectomy (PNX) study [112] revealed distinct chromatin accessibility profiles between Tom^High^ (AT2) and Tom^Low^ (AT2/AT1) cells. Tom^Low^ cells exhibited strong activation of FGFR2b signaling, as evidenced by upregulation of *Fgfr2b*, *Etv5*, *Sftpc*, *Ccnd1*, and *Ccnd2*, together with a trend towards higher *Ki67* expression, indicating the IAAPs’ phenotype [18], while Tom^High^ remained unresponsive under the same conditions. Transcriptomic profiling further identified immune-related surface molecules enriched in Tom^Low^, including *Cd33*, *Cd300lf,* and particularly *Cd274* (*PD-L1*). qPCR confirmed elevated expression of *Pd-l1* in Tom^Low^ relative to Tom^High^, and both immunofluorescence and flow cytometry demonstrated a high proportion of PD-L1-positive Tom^Low^ cells. Given the established role of PD-L1 as an immune checkpoint ligand and its prominent expression in lung adenocarcinoma, its enrichment in Tom^Low^ IAAPs [113,114,115,116] reinforces the concept that this population is uniquely equipped for immune regulation within the alveolar niche. Although such properties may be advantageous during injury and regeneration, they may also have implications in tumor biology. Future work should therefore investigate the dual role of these *PD-L1*^+^ IAAPs in repair, alveologenesis, and disease progression.

### 1.17. Killing 2 Birds with One Stone: Finding Antifibrotic and Pro-Regenerative Drugs Using the WI-38 Cell-Based Model

As outlined above, the WI-38-based system provides a robust platform for screening candidate anti-fibrotic and pro-regenerative compounds. When combined with the alveolosphere model, these approaches enable the evaluation of drug effects on both mesenchymal and epithelial counterparts, either independently or in conjunction, with the ultimate goal of driving disease regression and promoting de novo alveolar regeneration. FGF10 is indispensable for branching and alveolar morphogenesis [95]. In the therapeutic context, Ref. [30] demonstrated that exogenous rFGF10 exerts both protective and therapeutic effects against BLM-induced pulmonary fibrosis, providing fundamental preclinical support for its potential use in IPF. Consistent with the established role of FGF10/FGFR2b signaling in epithelial repair [18,29,112], rFGF10 not only limited fibrotic progression but also promoted alveolar regeneration, likely reflecting its combined impact on epithelial and mesenchymal compartments. AT2 injury through the deletion of *Fgfr2b* caused a marked loss of mature AT2 cells through apoptosis, whereas injury-activated alveolar progenitors (IAAPs) survived, expanded, and displayed enhanced regenerative capacity [29]. Importantly, in the study of [30], IAAPs were amplified in both *murine* models of BLM injury and in IPF patients, with their relative abundance peaking at the height of fibrosis when mature AT2s were most depleted. rFgf10 administration further prolonged and amplified the IAAP response, supporting the view that these progenitor-like cells, rather than canonical AT2s, may serve as the primary drivers of alveolar repair [30]. While the precise mechanisms remain to be fully resolved, these findings suggest that IAAPs may represent a key regenerative reserve in the injured lung. Targeting this population through rFgf10 holds promise for halting the progression of fibrosis and restoring alveolar architecture. In the context of the mesenchymal side, evidence shows that the intratracheal administration of Fgf10 mobilizes lung-resident-mesenchymal stem cells (LR-MSCs), which can be readily isolated by bronchoalveolar lavage and display classical MSC features and multipotency. These LR-MSCs, expanded by Fgf10, protect against diverse forms of lung injury and likely represent an organ-specific progenitor pool supporting epithelial repair [117]. Extending this concept, indications from our group demonstrate that *Gli1^+^* cells generate repair-supportive smooth muscle-like cells (RSMCs) after injury, where they acquire *Acta2* expression and accumulate in the peribronchial region. Crucially, these RSMCs serve as a privileged mesenchymal niche by producing FGF10, which sustains club cell progenitors and ensures effective epithelial regeneration, while loss of Fgf10 in *Gli1^+^* cells markedly impair repair [118]. Studying the WI-38 model in conjunction with FGF10 could provide important insights into how FGF10 contributes to both anti-fibrotic strategies and the promotion of alveolar regeneration.

Metformin, first highlighted by our group [41], has emerged as a key candidate in anti-fibrotic research; while clinical trials report conflicting effects on patient survival, they consistently suggest that metformin may confer benefits in IPF [119,120]. Importantly, the cohort study of COPD patients (GOLD 1–4; n = 1541; mean age 64.4 y; 601 females) showed that metformin therapy was associated with a significantly attenuated annual decline in pulmonary diffusing capacity (KCO: 0.2 vs. 2.3% predicted; TLCO: 0.8 vs. 2.8% predicted; *p* < 0.05) relative to non-diabetic control patients [121]. These findings support a potential role for metformin in mitigating emphysema-related alveolar dysfunction. Increasingly, this indicates that it may improve health status, symptoms, hospitalization rates, and mortality in those with Type 2 Diabetes Mellitus (T2DM), while no significant benefits were observed in patients without T2DM and not receiving metformin [122]. Recent studies using rat cigarette smoke-induced COPD and in vitro models demonstrated that metformin is protective by improving alveolar structure and lung function, mitigating inflammation and oxidative stress via activation of the Nrf2/HO-1/MRP1 pathway [123]. Elastase is commonly employed in animal models to mimic emphysematous changes in COPD by degrading elastin within the extracellular matrix, leading to alveolar destruction [124,125]. In a cellular context, the WI-38–alveolosphere system combined with AT2 cells can utilize elastase to model ECM remodeling and alveolar injury. This allowed for the investigation of AT2 responses to ECM stress and the supportive role of mesenchymal fibroblasts in tissue repair. Additionally, while elastase effectively models structural alveolar damage, it does not fully reproduce the chronic inflammatory components of COPD, which may require complementary stimuli such as cytokines [126] or cigarette smoke extract [124,125]. Nevertheless, this approach provides a valuable platform for studying alveolar injury and regeneration, particularly the interactions between AT2 cells and WI-38 fibroblasts, in a controlled, *human*-relevant system.

The emerging paradigm of metformin is particularly compelling, and when combined with the potential of testing novel targets in the WI-38 model, it may open new avenues for IPF-COPD research. Such approaches hold promise not only for halting disease progression but also for promoting tissue regeneration. TGF-β is a master regulator promoting both conditions [127,128]. Thus, it would be of high interest to test the potential effect of TGF-β inhibitors in the WI-38 model, together with alveolosphere, in the context of TGF-β-induced IPF or elastase-induced COPD, which could give some interesting findings.

Another characteristic shared by both diseases is their reliance on inflammatory responses [2,126]. Although several anti-inflammatory targets have not yielded promising outcomes in clinical trials [129], others remain under active investigation, as comprehensively reviewed by [2]. Consequently, the field remains open to testing the effects of diverse inflammatory inhibitors in cellular models, which may provide valuable insights.

Using a cellular approach, a recent study demonstrated that *mouse* fibroblasts can be directly converted into alveolar epithelial-like cells by introducing specific transcription factors, including *Nkx2-1*, *Foxa1*, *Foxa2*, and *Gata6,* in an in vitro system. These reprogrammed cells showed hallmarks of alveolar type 2 cells, including lamellar body-like structures and characteristic molecular markers. While their ability to differentiate into alveolar type 1 cells in vitro was limited, when delivered into the lungs of *mice* with bleomycin-induced fibrosis, the cells integrated into the alveolar tissue and gave rise to both type 1 and type 2-like alveolar cells [129]. This work presents a promising approach for generating functional alveolar cells from fibroblasts, paving the way for regenerative lung therapies. This also gives rise to the probability of using *human* cellular model reprogramming for a potential *human* approach.

These findings suggest that future therapeutic strategies should focus not only on slowing disease progression but also on actively promoting alveolar regeneration. Considering the complexity of IPF and COPD, a combined approach targeting multiple cellular and molecular pathways may be most effective (Table 2). For example, stimulating progenitor cells with FGF10 to support both epithelial and mesenchymal compartments, in conjunction with metformin to reduce inflammation and oxidative stress, could enhance repair. At the same time, modulating TGF-β signaling to limit fibrosis and testing anti-inflammatory interventions to refine the microenvironment may further support regeneration. Using the WI-38–alveolosphere co-culture system under TGF-β or elastase pressure to test such combinatorial strategies could provide new insights into cellular interactions and reveal therapeutic opportunities that have not yet been explored (Table 3).

## 2. Discussion

The dynamic interplay between alveolar epithelial type 2 (AT2) cells, lipofibroblasts (LIFs), and immune components during lung injury and repair represents a complex biological jigsaw. The emergence of AT2/AT1 intermediate states (variously termed PATS, ADIs, or AT0 cells) during alveolar repair presents a fascinating biological enigma [3,5,13,15,16,21]. These cells, characterized by the co-expression of *KRT8*, *CLDN4*, and *CTGF*, display remarkable phenotypic plasticity. While their persistence correlates with disease progression, their transient state during pathophysiology suggests an evolutionarily conserved role in alveolar regeneration. This notion raises important questions about the molecular switches that determine whether these cells progress to terminal AT1 differentiation or become arrested in an intermediate, disease-promoting state. The fibroblast–AT2 interaction in both homeostatic and disease states is well described [2,29,37,40,130,131,132]. However, the reciprocal signaling from activated myofibroblasts that affects the AT2/AT1 intermediate states remains incompletely characterized. Here, the WI-38–alveolosphere co-culture system provides a compelling *human*-relevant platform to interrogate bidirectional communication between alveolar epithelial and mesenchymal compartments. Such a platform is essential not only for understanding alveolar injury and repair but also for defining interventions that can promote true alveologenesis—a goal that represents the ultimate therapeutic win against IPF and COPD (Table 4).

The identification of haloperidol as an inhibitor of myofibroblast activation expands the potential pharmacological arsenal against fibrosis and chronic pulmonary conditions [43], suggesting an effective role for neuromodulation, which has received limited attention until now. In this context, the WI-38–alveolosphere assay incorporating neuromodulators or even neuronal cells would be of high interest. Similarly, the miR-29 family’s ability to suppress TGF-β signaling and promote AT2 differentiation highlights the potential of microRNA-based therapies for alveolar regeneration.

The clinical solution likely lies in precise, context-dependent interventions: simultaneously putting the brakes on destructive AREG signaling and stalled AT2/AT1 differentiation, while accelerating the pro-regenerative pathways, such as FGF10 and YAP/TAZ signaling. Notably, emerging evidence suggests these transitional cells may not be permanently lost to disease. Even in advanced fibrosis or expanding to COPD [133,134,135,136], lineage tracing reveals AT2/AT1 intermediates that, through interventions, maintain their capacity to contribute to alveolar repair [6]. This could be a clinical game-changer. A treatment paradigm must evolve beyond simply stopping chronic injury to actively jumpstart the lung’s regenerative programs and promote alveolar regeneration.

FGF10 emerges as a pivotal factor in this context. Essential for branching morphogenesis and alveolar development [95], Fgf10 promotes epithelial repair and alveolar regeneration in preclinical models of fibrosis [18,29,30,112]. Notably, injury-activated alveolar progenitors (IAAPs) survive AT2 injury, expand, and are amplified further upon rFgf10 administration, supporting the view that these progenitor-like cells drive alveolar repair. On the mesenchymal side, Fgf10 mobilizes lung-resident mesenchymal stem cells (LR-MSCs) and repair-supportive *Gli1^+^* cells, which serve as a niche that produces Fgf10 and sustains epithelial regeneration [117,118]. Together, these findings highlight the therapeutic potential of targeting both epithelial and mesenchymal compartments through FGF10-based interventions.

Metformin similarly holds promise, with emerging evidence suggesting that it attenuates emphysema-related alveolar dysfunction and improves outcomes in COPD patients with Type 2 diabetes mellitus [121,122,123]. Using WI-38–alveolosphere systems in combination with elastase or TGF-β injury models allows for the study of ECM remodeling, alveolar damage, and the supportive role of mesenchymal fibroblasts under controlled *human*-relevant conditions. Such combinatorial platforms can dissect mechanisms of injury and regeneration, providing actionable insights for therapeutic development.

Beyond these canonical pathways, the pharmacological landscape expands further. A mentioned above haloperidol inhibits myofibroblast activation [43], and miR-29 family members suppress TGF-β signaling while promoting AT2 differentiation, highlighting the potential of neuromodulation and microRNA-based therapies. The convergence of these approaches underscores that precise, context-dependent interventions simultaneously limiting destructive signals, such as AREG or TGF-β, and promoting pro-regenerative pathways, like FGF10, YAP/TAZ, or Wnt/β-catenin signaling, may be required to restore alveolar architecture.

Importantly, even in advanced fibrosis or COPD, lineage tracing reveals that AT2/AT1 intermediates retain regenerative potential [6,133,134,135,136]. This suggests that stage-specific, temporally controlled interventions such as early disruption of AREG/EGFR signaling and late-stage promotion of AT1 maturation via YAP/TAZ or Wnt modulation could harness the lung’s intrinsic regenerative programs. Combinatorial strategies addressing both epithelial dysfunction (via FGFR2b agonism or IL-1β inhibition) [137,138] and mesenchymal activation (via TGF-β blockade or PPARγ-mediated LIF reprogramming) are particularly promising.

Lung injury resolution is a gradual, multifactorial process that depends on synchronized epithelial and mesenchymal remodeling rather than a simple reversal of fibrosis. During this phase, the restoration of alveolar architecture depends on the capacity of surviving AT2 cells to re-enter the cell cycle, self-renew, and differentiate into mature AT1 cells through transitional intermediates such as DATPs and PATS. In parallel, activated myofibroblasts progressively lose their contractile phenotype and reacquire lipogenic characteristics typical of quiescent lipofibroblasts. This phenotypic reversion is supported by PPARγ activation and by the re-establishment of paracrine signaling networks involving FGF10, Wnt, and temporally regulated AREG–EGFR activity. The balance between pro-fibrotic and pro-regenerative cues determines whether the lung proceeds toward structural restitution or persistent remodeling. Consequently, impaired resolution is often associated with sustained inflammatory signaling, accumulation of transitional epithelial states, and failure of fibroblast deactivation. Understanding the molecular checkpoints that govern this transition offers new opportunities to promote endogenous repair and prevent chronic fibrosis.

Ultimately, the development of *human*-relevant platforms such as the WI-38–alveolosphere system is crucial for testing these multiphasic, context-specific therapies. Such systems allow for the interrogation of progenitor cell behavior, epithelial–mesenchymal crosstalk, and the dynamic cellular transitions that underpin alveologenesis. Future studies should prioritize validating interventions in vivo to bridge current in vitro findings with clinically relevant outcomes. By enabling the precise modeling of both IPF and COPD pathophysiology, these platforms provide an invaluable tool for advancing therapies that not only slow disease progression but also actively restore lung architecture, offering the potential for true alveolar regeneration.

## 3. Conclusions

Despite rapid advances in single-cell mapping and lineage-tracing technologies, several critical knowledge gaps remain. The precise molecular cues that govern the transition between fibroblast states, the spectrum of AT2 transitional intermediates in *human* lungs, and the determinants of epithelial–mesenchymal reciprocity still remain to be understood. Moreover, most mechanistic insights derive from *murine* injury models, highlighting the need for validation in *human* tissue and ex vivo platforms. From a translational standpoint, integrating in vitro models such as WI-38 fibroblast–alveolosphere co-cultures and precision-cut lung slices could accelerate the identification of context-specific antifibrotic therapies. Clarifying these regulatory checkpoints will be essential to bridge fundamental discovery with clinical intervention in chronic lung diseases such as IPF and COPD.

This review underscores the dynamic reciprocity between alveolar epithelial type 2 (AT2) cells and lipofibroblasts (LIFs) as the central determinant of lung injury resolution or progression toward fibrosis. Transitional AT2/AT1 states, coupled with mesenchymal activation and AREG–EGFR signaling, define a reversible but tightly regulated network that dictates alveolar fate. Integrating *human*-relevant systems such as WI-38 fibroblast and alveolosphere co-culture models provides a practical framework for dissecting these epithelial–mesenchymal interactions and testing pharmacological interventions. Compounds such as metformin, haloperidol, rosiglitazone, saracatinib, and FGF10 exemplify strategies capable of reprogramming fibrotic pathways while enhancing epithelial regeneration. Together, these advances highlight the feasibility of a dual therapeutic paradigm—simultaneously halting fibrosis and restoring alveolar architecture—paving the way for truly regenerative treatments in idiopathic pulmonary fibrosis and COPD.

## Figures and Tables

**Figure 2 cells-14-01869-f002:**
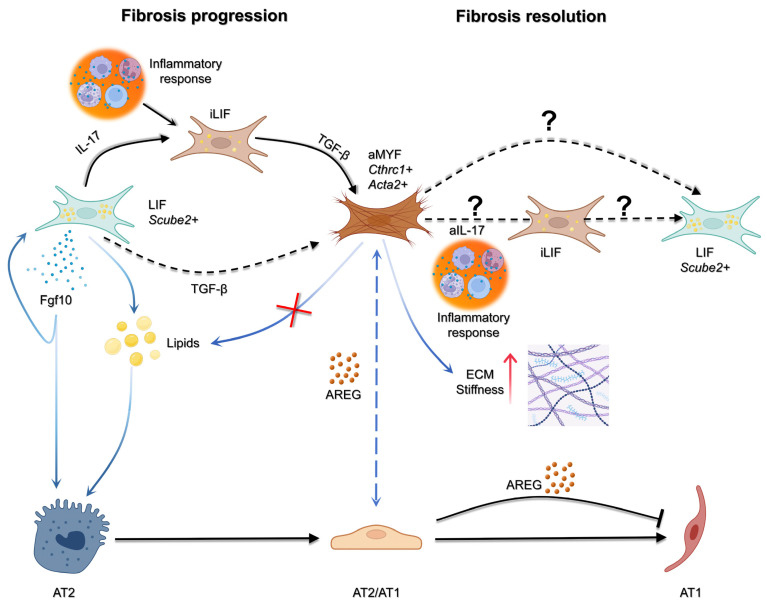
Cellular interactions driving fibrosis progression and resolution. During injury, inflammatory cues such as IL-17 promote the emergence of inflammatory LIF+ fibroblasts (iLIF), which in turn release TGF-β to drive activation of myofibroblasts (aMYF; Cthrc1+ Acta2+). Activated fibroblasts reinforce fibrosis in the epithelial side through AREG stalling the epithelium in an intermediate state. In contrast, LIF+ fibroblasts (Scube2+) can provide pro-regenerative signals, including Fgf10 and lipids, that support alveolar type 2 (AT2) cell survival and differentiation into alveolar type 1 (AT1) cells. The balance between these pathways determines outcome: persistence of aMYF activity promotes fibrosis progression, while the potential reversion or remodeling of fibroblast states (aIL-17, iLIF, or LIF+ populations) may underlie fibrosis resolution. To date, no evidence indicate the reversion of aMYF to LIF through iLIF and it would be of interest to be studied. Figure created with BioRender.com.

**Figure 3 cells-14-01869-f003:**
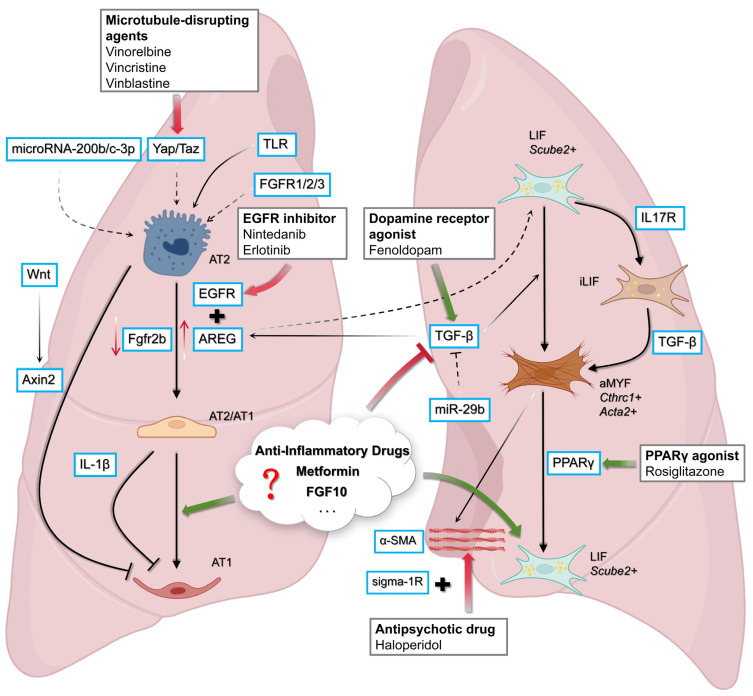
Therapeutic targets and signaling pathways in alveolar repair and fibrosis. Multiple pathways regulate the balance between alveolar epithelial regeneration and fibrotic remodeling. In AT2 cells, Wnt/β-catenin signaling and EGFR/AREG activity promote differentiation towards complete AT1 or AT2/AT1, respectively. EGFR inhibitors (e.g., nintedanib, erlotinib) and dopamine receptor agonists (e.g., fenoldopam) can attenuate profibrotic signaling, whereas microtubule-disrupting agents and miR-29b act to suppress TGF-β activity. On the mesenchymal side, LIF fibroblasts (Scube2+) provide regenerative support, counterbalanced by inflammatory fibroblast states (iLIF, aMYF) that drive TGF-β-dependent matrix deposition and stiffness. Pharmacological interventions, including PPARγ agonists (rosiglitazone), antipsychotics (haloperidol, via sigma-1R), and potentially other agents such as anti-inflammatory compounds, Metformin, and FGF10, are poised to tip the balance toward epithelial repair and fibrosis resolution. Figure created with BioRender.com.

**Table 2 cells-14-01869-t002:** Conceptual framework summarizing the sequential cellular and molecular interactions between alveolar epithelial and mesenchymal compartments during injury, fibrosis, and repair. This table integrates findings discussed in the main text and should be interpreted as a schematic synthesis rather than direct experimental evidence.

Disease Stage	Key Pathobiology	Therapeutic Logic	Example Interventions
Early Injury	AT2 injury, DAMP release (IL-1β, TNFα) AREG–EGFR signaling stalls AT2/AT1 intermediates	Block excessive inflammation and EGFR-driven stalling while preserving AT2 pool	Targeted anti-inflammatory modulation (IL-1β, TNFα inhibitors) EGFR/AREG blockade (nintedanib, erlotinib)
Intermediate	Transitional AT2/AT1 states persistence Fibroblasts: LIF → iLIF → aMYF	Promote resolution & regenerative programs in both epithelium and mesenchyme	PPARγ agonists (rosiglitazone, metformin) for fibroblast reprogramming FGF10, YAP/TAZ activation, microtubule disruptors (vinorelbine, mebendazole) for epithelial repair miRNAs (miR-29b, miR-200 family) to restore AT2 → AT1 differentiation
Late Fibrosis	Persistent aMYFs, ECM deposition Loss of alveolar structure, impaired regeneration	Actively reverse fibrosis and mobilize regenerative reserves	Anti-fibrotics (TGF-β/ALK5 inhibitors, saracatinib) Neuromodulatory antifibrotics (haloperidol, dopamine receptor agonists) Fibroblast lipogenic reprogramming (PPARγ agonists, metformin) rFGF10 to reverse alveolar epithelial intermediates

**Table 3 cells-14-01869-t003:** Proposed pipeline for therapeutic discovery in lung injury and fibrosis. The workflow begins with fibroblast state manipulation (LIF/iLIF/aMYF balance), proceeds through alveolosphere co-culture readouts, integrates multi-omics profiling to dissect epithelial–mesenchymal interaction, and results in therapeutic prioritization. This pipeline is designed as a conceptual framework; interventions highlighted are hypothetical and exploratory.

Proposed Hypothetical Pipeline for Therapeutic Discovery	
**1. Fibroblast State Manipulation**	Use WI-38 fibroblasts as a manipulable system. Drive fibroblast states: LIF → iLIF → aMYF (via metformin, IL-17, TGF-β). Model reversibility: promote reprogramming back toward lipogenic LIF.
**2. Alveolosphere Co-culture Readouts**	Combine WI-38 fibroblast states with AT2 cells. Measure epithelial outcomes: AT2→AT1 differentiation, intermediate state persistence. Readouts: organoid size, alveolar-forming efficiency, immunofluorescence, RT-qPCR.
**3. Multi-omics Profiling**	Perform bulk RNA-seq, scRNA-seq, proteomics. Species-separated readouts (*mouse* epithelium vs. *human* fibroblast). Identify epithelial–mesenchymal signaling pathways (AREG/EGFR, TGF-β, FGF10, YAP/TAZ, miRNAs).
**4. Therapeutic Prioritization**	Rank drug candidates (e.g., metformin, PPARγ agonists, haloperidol, miRNAs, saracatinib, rFGF10). Prioritize interventions that promote AT2→AT1 maturation, reverse fibroblast activation, and enhance alveolosphere regeneration. Feedback into successful testing in alveolosphere + WI-38 models. Advance to in vivo model, and clinical trial.

**Table 4 cells-14-01869-t004:** Comparative overview of therapeutic strategies targeting epithelial–mesenchymal signaling pathways in lung injury and fibrosis.

Target Pathway	Cellular Target	Mechanistic Effect	Representative Agents	Therapeutic Outcome	References
AREG–EGFR	AT2 cells/fibroblasts	Inhibition of profibrotic EGFR signaling; release of stalled AT2/AT1 intermediates	Nintedanib, Erlotinib	Restores AT2→AT1 differentiation and reduces fibrosis	[6,52,69]
TGF-β/SMAD2/3	Fibroblasts/AT2 cells	Blockade of myofibroblast activation and epithelial–mesenchymal transition (EMT)	ALK5 inhibitors, Saracatinib, miR-29b	Attenuates ECM deposition and promotes epithelial regeneration	[47,51,84,93]
FGF10–FGFR2b	AT2 cells/mesenchyme	Activation of epithelial survival and progenitor renewal pathways	Recombinant FGF10 (rFGF10)	Enhances alveolar repair	[18,29,30,112]
PPARγ Activation	Fibroblasts	Induction of lipogenic reprogramming; reversal of activated myofibroblast (aMYF) phenotype	Rosiglitazone, Metformin	Promotes fibrosis resolution and restoration of LIF phenotype	[36,40,41,45]
YAP/TAZ (Hippo Signaling)	AT2 cells	Activation of regenerative and proliferative responses post-injury	Microtubule disruptors (vinorelbine, mebendazole, colchicine)	Stimulates AT2 proliferation and regeneration	[58,59]
Wnt/β-Catenin	AT2 cells	Regulation of differentiation plasticity through Axin2-dependent signaling	Tankyrase inhibitors, Wnt modulators	Promotes AT2→AT1 differentiation	[55]
microRNAs	AT2 cells	Restoration of differentiation balance; inhibition of fibrotic signaling	miR-29 family, miR-200 family	Reverses EMT and promotes AT1 maturation	[84,89]
Neuromodulatory Signaling	Fibroblasts/AT2 cells	Modulation of sigma-1R and dopamine receptor pathways to limit myofibroblast activation	Haloperidol, Fenoldopam	Reduces fibroblast activation and enhances tissue repair	[43,44]

## Data Availability

No new data were created or analyzed in this study.

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
