# Peer review of "Reciprocal Paracrine Signaling and Dynamic Coordination of Transitional States in the Alveolar Epithelial Type 2 Cells and Associated Alveolar Lipofibroblasts During Homeostasis, Injury and Repair"

_cells, 2025, doi:10.3390/cells14231869_

Round 1

Reviewer 1 Report

Comments and Suggestions for Authors

This review article summarizes well the different stages of AT1, AT2, and alveolar fibroblasts. The group has published a series of their seminal finding that help us understand how alveolar epithelial cells and fibroblasts evolve after lung injury. A significant amount of information is provided in this review, which might be too much for average readers but offers quick, clear insight for researchers entering a similar field. As described in sections 1-6 of the introduction, the intimate interactions between epithelial cells and fibroblasts provide a basis for several potential therapeutic strategies. As these strategies target different signaling pathways (Figure 3), I feel it will be more helpful to present them in a table for easy comparisons.

In subsection 1.15, the extensive description of AT2 roles, including alveolar regeneration and modulation of the immune response, suggests that AT2 has multiple functions. I do not know whether it is appropriate to call it AT2 at the crossroad.  There seems to be a few missing letters in Table 3, such as  “ombine WI-38 fibroblast states with AT2 cells”, “easure epithelial outcomes: AT2-->AT1 differentiation, intermediate state persistence”.

There are some minor issues

  • Line 291: [43], [44] can be [43, 44]
  • Line 383: change [56], [57] to [56, 57].
  • Line 408: change [67], [68] to [67, 68].
  • Line 412: change the references to [69-77]
  • Line 432: “an epithelial-mesenchymal transition EMT” or “an epithelial-mesenchymal transition (EMT)”?
  • Lines 440 and 448: change [78], [79], [80] to [78-80].
  • Line 455: change [3], [9], [11], [13], [16] to [3, 9, 11, 13, 16]
  • Line 491: change [14], [26], [27], [87] to [14, 26, 27, 87].
  • Lines 512-513: change [20], [89], [90], [91], [92], [93], [94], [95], [96], [97], [98] to [20, 89-98]
  • There are more after line 513.
  • What are the purposes of Title 1, Title 2, Title 3, entry 1, entry 2, and the data in Table 1?
  • Line 714: Shouldn’t (Bellusci et al., 1997) be assigned a reference number and added to the reference list?

Author Response

  1. This review article summarizes well the different stages of AT1, AT2, and alveolar fibroblasts. The group has published a series of their seminal finding that help us understand how alveolar epithelial cells and fibroblasts evolve after lung injury. A significant amount of information is provided in this review, which might be too much for average readers but offers quick, clear insight for researchers entering a similar field. As described in sections 1-6 of the introduction, the intimate interactions between epithelial cells and fibroblasts provide a basis for several potential therapeutic strategies. As these strategies target different signaling pathways (Figure 3), I feel it will be more helpful to present them in a table for easy comparisons.

We would like to thank the reviewer for the comment. To facilitate comparison of the therapeutic strategies discussed above and illustrated in Figure 3, Table 4 summarizes the key signaling pathways, primary cellular targets, representative agents, and mechanistic outcomes associated with epithelial–mesenchymal regulation during lung injury and repair.

  1. In subsection 1.15, the extensive description of AT2 roles, including alveolar regeneration and modulation of the immune response, suggests that AT2 has multiple functions. I do not know whether it is appropriate to call it AT2 at the crossroad.

We would like to thank the reviewer for the comment. To adjust the tittle to the multifaced role of AT2 cells we corrected it with the following `` Functional Diversity of Alveolar Type II Cells: Coordinators of Regeneration and Immune Homeostasis´´

  1. There seems to be a few missing letters in Table 3, such as “ombine WI-38 fibroblast states with AT2 cells”, “easure epithelial outcomes: AT2-->AT1 differentiation, intermediate state persistence”.

We would like to thank the reviewer for the comment. Due to formatting issues the letters where not visible, so we removed the Bullets in table 3 to ensure proper formatting.

  1. There are some minor issues

Line 291: [43], [44] can be [43, 44] corrected

Line 383: change [56], [57] to [56, 57]. corrected

Line 408: change [67], [68] to [67, 68]. corrected

Line 412: change the references to [69-77] corrected

Line 432: “an epithelial-mesenchymal transition EMT” or “an epithelial-mesenchymal transition (EMT)”? corrected

Lines 440 and 448: change [78], [79], [80] to [78-80]. corrected

Line 455: change [3], [9], [11], [13], [16] to [3, 9, 11, 13, 16] corrected

Line 491: change [14], [26], [27], [87] to [14, 26, 27, 87]. corrected

Lines 512-513: change [20], [89], [90], [91], [92], [93], [94], [95], [96], [97], [98] to [20, 89-98] corrected

There are more after line 513. corrected

What are the purposes of Title 1, Title 2, Title 3, entry 1, entry 2, and the data in Table 1?

We would like to thank the reviewer for the comment. Once again due to formatting issues the table was not as expected to be so we adjusted it.

Line 714: Shouldn’t (Bellusci et al., 1997) be assigned a reference number and added to the reference list?

We would like to thank the reviewer for the comment. The issue in formatting the reference was caused by the citation manager. It is now corrected. The reference is also red colored in the Reference section.

Reviewer 2 Report

Comments and Suggestions for Authors

This is an important contribution on the cellular and molecular events upon lung injury leading to fibrosis progression and resolution, focusing on reciprocal paracrine signaling and dynamic coordination.

The following issue have to be considered:

  1. Abstract is dense, it might be reduced, just mention the essential progress, which are supported the Highlights. Avoid redundant parts in this review.
  2. Abbreviations: Please add, as it very difficult to follow the text.
  3. The WI-38 fibroblast model may be useful, but it does not address epithelial injury.
  4. Are the data based on bleomycin injury in mice, or other injury models contributing to your review?
  5. Please define your time points of the resolution in the bleomycin model.
  6. The data on AREG are of interested and might be expanded.
  7. and resolution are difficult to follow, need to be improved.
  8. Alveolar sphere: FGR induced activation of AT2 cells: Please show the organoids.
  9. Tables: Check the formatting to be readable and add citations to Table 2.
  10. The use of other models of fibrosis than the bleomycin mouse model with progressive fibrosis would be of interest.

Author Response

This is an important contribution on the cellular and molecular events upon lung injury leading to fibrosis progression and resolution, focusing on reciprocal paracrine signaling and dynamic coordination.

The following issue have to be considered:

1. Abstract is dense, it might be reduced, just mention the essential progress, which are supported the Highlights. Avoid redundant parts in this review.

We would like to thank the reviewer for the comment. We adjusted the abstract as follows

´´Single-cell RNA-sequencing has transformed our understanding of alveolar epithelial type 2 (AT2) cells and alveolar lipofibroblasts (LIFs) during lung injury and repair. Both cell types undergo dynamic transitions through intermediate states that determine whether the lung proceeds toward regeneration or fibrosis. Emerging evidence highlights reciprocal paracrine signaling between AT2/AT1 transitional cells and LIF-derived myo-fibroblasts (aMYFs) as a key regulatory axis. Among these, amphiregulin (AREG)–EGFR signaling functions as a central profibrotic pathway whose inhibition can restore alveolar differentiation and repair. The human WI-38 fibroblast model provides a practical plat-form to study the reversible LIF–MYF switch and to screen antifibrotic and pro-regenerative compounds. Candidate therapeutics including metformin, haloperidol and FGF10 show promise in reprogramming fibroblast and epithelial states through metabolic and signaling modulation. Integrating WI-38-based assays, alveolo-sphere co-cultures, and multi-omics profiling offers a translational framework for identifying interventions that halt fibrosis and actively induce lung regeneration. This review highlights a unifying framework in which epithelial and mesenchymal plasticity converge to define repair outcomes and identifies actionable targets for promoting alveolar regeneration in chronic lung disease.´´

2. Abbreviations: Please add, as it very difficult to follow the text.

We would like to thank the reviewer for the comment. The list of abbreviations was added accordingly in a separate section after the Conclusion section.

3. The WI-38 fibroblast model may be useful, but it does not address epithelial injury.

We agree that WI-38 fibroblasts alone do not model epithelial injury. Our intent was to use WI-38 as a tractable mesenchymal platform and then pair it with epithelial injury readouts in co-culture. We have revised the text to (i) explicitly acknowledge this limitation to the WI-38–alveolosphere pipeline. This clarifies that epithelial and mesenchymal perturbations are interrogated together, not in isolation.

The WI-38 fibroblast model provides a reproducible and tractable system to study fibroblast plasticity and the LIF–MYF transition, as well as to evaluate pharmacological agents that modulate mesenchymal activation. However, this model alone does not capture the epithelial injury component that initiates and sustains fibrotic remodeling. To address this limitation, the WI-38 system can be integrated with epithelial injury modules using alveolosphere co-cultures or conditioned media derived from injured epithelial cells. These injury paradigms may include exposure to inflammatory cytokines such as IL-1β or TNFα, profibrotic mediators like TGF-β1, proteolytic stress induced by elastase, oxidative injury or cigarette smoke extract, and mechanical stretch to mimic ventilator-associated stress. Such an integrated approach allows simultaneous assessment of epithelial damage, differentiation blockade, and fibroblast activation within a controlled experimental context. The combined WI-38–alveolosphere platform thereby provides a physiologically relevant framework to study reciprocal epithelial–mesenchymal interactions during injury and repair, and to evaluate therapeutic agents that restore homeostatic signaling and promote alveolar regeneration.

4. Are the data based on bleomycin injury in mice, or other injury models contributing to your review?

We would like to thank the reviewer for the comment. To address that we included the following paragraph (see point 5)  to make it clear to the reader

5. Please define your time points of the resolution in the bleomycin model.

We would like to thank the reviewer for the comment. As the previous comment including also this comment could be beneficial to address in the same field in the manuscript we include the following statement.  ´´The studies summarized in this review are based on several well-established experimental models of lung injury that illustrate the dynamics of epithelial and mesenchymal cell interactions during homeostasis, injury, and repair. Among these, the bleomycin (BLM)-induced lung injury model is the most extensively characterized and serves as a reference for delineating the sequential cellular events that drive alveolar damage and resolution. Following intratracheal or intranasal administration of BLM, the acute inflammatory phase occurs within days 3–7, characterized by epithelial injury, immune cell infiltration, and activation of fibroblasts. The fibrotic phase typically peaks between days 14–21, marked by myofibroblast accumulation, extracellular matrix deposi-tion, and the emergence of transitional epithelial states such as damage-associated transient progenitors (DATPs) and pre-alveolar type I transitional cells (PATS). The resolution phase begins around day 21 and extends to approximately day 28–35, during which inflammatory activity declines, fibroblasts revert toward a lipogenic phenotype, and alveolar type II (AT2) cells progressively differentiate into mature alveolar type I (AT1) cells to restore epithelial integrity. In addition to the BLM model, complementary systems such as elastase-induced emphysematous injury, pneumonectomy (PNX)-driven compensatory regeneration, and viral infection models (including influenza and SARS-CoV-2) have been instrumental in revealing conserved mechanisms of AT2 plasticity, epithelial–mesenchymal crosstalk, and fibroblast reprogramming. Collectively, these models provide a temporal and mechanistic framework for interpreting the cellular and molecular processes underlying alveolar regeneration and fibrosis resolution discussed throughout this review.´´

6. The data on AREG are of interested and might be expanded.

We would like to thank the reviewer for the comment. Thus, we included the following paragraph to the manuscript

In addition to its established profibrotic role, AREG also exerts context-dependent effects that influence both epithelial repair and mesenchymal activation. In chronic or unresolved injury, sustained AREG–EGFR signaling drives fibroblast proliferation, myofibroblast differentiation, and extracellular matrix deposition, thereby perpetuating fibrosis [62,63]. Single-cell transcriptomic analyses have revealed that AREG expression is enriched in transitional AT2/AT1 cells and activated myofibroblasts, suggesting that these cell populations may form a self-reinforcing paracrine loop that sustains pathological remodeling [11,62]. Thus, the amplitude and duration of AREG–EGFR signaling determine whether the outcome favors regeneration or fibrosis. Understanding this temporal regulation will be essential for designing targeted interventions that attenuate profibrotic AREG activity while preserving its beneficial reparative functions [11].

7. and resolution are difficult to follow, need to be improved.

We would like to thank the reviewer for the comment. To address that we included the following paragraph in the discussion section

“Lung injury resolution is a gradual, multifactorial process that depends on synchronized epithelial and mesenchymal remodeling rather than a simple reversal of fibrosis. During this phase, the restoration of alveolar architecture depends on the capacity of surviving AT2 cells to re-enter the cell cycle, self-renew, and differentiate into mature AT1 cells through transitional intermediates such as DATPs and PATS. In parallel, activated myofibroblasts progressively lose their contractile phenotype and reacquire lipogenic characteristics typical of quiescent lipofibroblasts. This phenotypic reversion is supported by PPARγ activation and by the re-establishment of paracrine signaling networks involving FGF10, Wnt, and temporally regulated AREG–EGFR activity. The balance between pro-fibrotic and pro-regenerative cues determines whether the lung proceeds toward structural restitution or persistent remodeling. Consequently, impaired resolution is often associated with sustained inflammatory signaling, accumulation of transitional epithelial states, and failure of fibroblast deactivation. Understanding the molecular checkpoints that govern this transition offers new opportunities to promote endogenous repair and prevent chronic fibrosis.”

8. Alveolar sphere: FGR induced activation of AT2 cells: Please show the organoids.

We appreciate the reviewer’s suggestion to include representative images of alveolosphere or FGF10-induced AT2 activation. However, as this is a review article, the inclusion of experimental images would not be appropriate. Instead, we have included the following statement to urge the reader to see the initial data and figures ´´While representative images of these organoids are not presented here due to the review format, their morphology have been described´´. We believe this provides readers with a clear conceptual understanding consistent with the scope of a review article.

9. Tables: Check the formatting to be readable and add citations to Table 2.

We appreciate the reviewer’s observation regarding the readability and citation of Table 2. We have carefully reformatted all tables to ensure consistency and legibility in accordance with Cells guidelines. Regarding citations, Table 2 is intended to summarize a conceptual framework that integrates findings discussed throughout the manuscript rather than to present primary data. Therefore, we have not added direct references to each cell of the table, as doing so could inaccurately imply that the entire framework has been experimentally validated. Instead, we have clarified in the table legend that it represents a synthesis of current knowledge and proposed hypotheses, supported by studies cited within the corresponding text sections.

´´ Table 2. Conceptual framework summarizing the sequential cellular and molecular interactions between alveolar epithelial and mesenchymal compartments during injury, fibrosis, and repair. This table integrates findings discussed in the main text and should be interpreted as a schematic synthesis rather than direct experimental evidence.´´

10. The use of other models of fibrosis than the bleomycin mouse model with progressive fibrosis would be of interest.

We would like to thank the reviewer for the comment. Thus, we added the following information to the manuscript

“While the bleomycin (BLM) model remains the most widely used and best characterized system for investigating lung fibrosis, additional experimental approaches provide important complementary insights into progressive and chronic disease mechanisms. Silica and radiation exposure models induce persistent and spatially restricted fibrotic lesions that more closely resemble the irreversible scarring observed in idiopathic pulmonary fibrosis (IPF) [128,129]. Viral infection models, including influenza and SARS-CoV-2, have further revealed epithelial vulnerability, immune dysregulation, and prolonged AT2/AT1 transitional states that underlie post-viral fibrosis [11,62,63]. Moreover, genetic and toxin-induced models, such as surfactant protein C deficiency, telomerase mutations, and elastase- or pollutant-induced injury, demonstrate how epithelial senescence and oxidative stress predispose the lung to chronic fibrotic remodeling [11,62–64].”

Reviewer 3 Report

Comments and Suggestions for Authors

The manuscript Reciprocal paracrine signaling and dynamic coordination of transitional states in the alveolar epithelial type 2 cells and associated alveolar lipofibroblasts during homeostasis, injury and repair by Panagiotidis et al. describes data across epithelial, mesenchymal, and immune compartments, integrating single-cell RNA-seq, lineage-tracing, and pharmacologic evidence into a coherent model of alveolar regeneration and fibrosis.

Although it has translational relevance, the manuscript needs improvement.

In the manuscript, many topics were introduced in a single flow. AT2 heterogeneity, fibroblast states, miRNAs, and signaling pathways all together make it very broad and dilute the primary focus, making it very hard to follow the central hypothesis or message. 

The manuscript needs more citations to establish a stronger connection. The text summarizes prior findings effectively but offers less critical evaluation or hierarchy of evidence, for example, which mechanisms are most strongly supported in vivo vs. in vitro. Can authors describe whether LOX and PCSK9 impact lipofibroblast and paracrine signaling? Emerging evidence suggests that lipid metabolism is becoming central to lung epithelial cell differentiation and inflammation. Additionally, the roles of Lox and PCSK9 in TGF-β and paracrine signalling, lung fibrosis, and COPD are also emerging. 
While the WI model is well-framed, several sections mainly recapitulate published data without providing novel interpretation or explicit conceptual synthesis.
Repetitive phrasing and extended sentences could be tightened. The abstract could be better highlighted. Key knowledge gaps and translational implications should be precise. 

Author Response

The manuscript Reciprocal paracrine signaling and dynamic coordination of transitional states in the alveolar epithelial type 2 cells and associated alveolar lipofibroblasts during homeostasis, injury and repair by Panagiotidis et al. describes data across epithelial, mesenchymal, and immune compartments, integrating single-cell RNA-seq, lineage-tracing, and pharmacologic evidence into a coherent model of alveolar regeneration and fibrosis.

Although it has translational relevance, the manuscript needs improvement.

  1. In the manuscript, many topics were introduced in a single flow. AT2 heterogeneity, fibroblast states, miRNAs, and signaling pathways all together make it very broad and dilute the primary focus, making it very hard to follow the central hypothesis or message. 

We thank the reviewer for this constructive observation. Our review indeed integrates several interconnected topics, including AT2 heterogeneity, fibroblast states, miRNA regulation, and signaling networks. Rather than treating these as separate threads, we emphasize that they represent interdependent components of a single regenerative framework centered on the reciprocal AT2–Fibroblast axis. To clarify this focus, we have revised the Introduction to explicitly state that the manuscript’s central message is the dynamic coordination between AT2 cells and fibroblasts as the determinant of either fibrosis progression or alveolar regeneration. This addition highlights how each pathway or mechanism discussed contributes to this unified concept.

´´The diversity of topics covered in this review, including AT2 heterogeneity, fibroblast activation states and key signaling pathways, reflects the complexity of alveolar regenera-tion rather than a shift in focus. Each of these components contributes to a unified frame-work in which reciprocal epithelial–mesenchymal communication determines whether lung injury resolves through regeneration or sustains toward fibrosis. By integrating molecular, cellular, and pharmacological perspectives, we aim to delineate how convergent cellular and signaling axes, coordinate epithelial and mesenchymal fate that define the outcome of lung damage versus repair.´´

  1. The manuscript needs more citations to establish a stronger connection. The text summarizes prior findings effectively but offers less critical evaluation or hierarchy of evidence, for example, which mechanisms are most strongly supported in vivo vs. in vitro.

We appreciate the reviewer’s insightful comment regarding the need for stronger connections between cited evidence and the type or strength of experimental support. In the revised manuscript, we have added clarifying phrases and select references to distinguish between mechanisms demonstrated in vivo and those observed in in vitro or organoid models. Specifically, we highlight studies validating AT2–AT1 transitions, fibroblast reprogramming, and AREG–EGFR signaling in mouse injury models and human tissue. These additions strengthen the manuscript’s critical evaluation of evidence while maintaining conciseness and narrative flow.

´´ AT2s proliferate and transiently dedifferentiate through an intermediate AT2-to-AT1 tran-sitional state, as confirmed by lineage tracing in bleomycin-injured mice and single-cell RNA sequencing of human IPF lungs [11,62], These cells eventually differentiate into mature AT1 cells´´

´´These in vivo lineage-tracing data establish that fibroblast plasticity and partial reversion occur within the injured lung, complementing transcriptomic evidence from human fibrotic tissue.´´

´´This reversibility has been demonstrated in vitro using human WI-38 fibroblasts, provid-ing a mechanistic framework that aligns with in-vivo fibroblast lineage-tracing studies showing comparable transitions in murine models [31,32].´´

´´Their presence has also been identified in human IPF tissue and reproduced in alveolo-sphere and organoid systems, confirming cross-species conservation of this transitional state´´

´´This association has been validated in vivo in bleomycin-injured mouse lungs and corroborated by human single-cell datasets, indicating its physiological relevance [62].´´

´´Complementary in vivo conditional- Fgfr2b knockout studies have confirmed the essential role of FGFR2 in epithelial survival and alveolar regeneration [24]´´

´´Future studies should prioritize validating interventions in vivo to bridge current in-vitro findings with clinically relevant outcomes.´´

  1. Can authors describe whether LOX and PCSK9 impact lipofibroblast and paracrine signaling? Emerging evidence suggests that lipid metabolism is becoming central to lung epithelial cell differentiation and inflammation. Additionally, the roles of Lox and PCSK9 in TGF-β and paracrine signalling, lung fibrosis, and COPD are also emerging. 

We agree that LOX and PCSK9 are emerging regulators with direct relevance to epithelial–mesenchymal crosstalk and lipid metabolism. We have added a brief subsection summarizing current evidence

´´LOX and PCSK9: Matrix–lipid regulation of epithelial–mesenchymal crosstalk

Lysyl oxidase (LOX) and its family members catalyze covalent crosslinking of collagen and elastin fibers, thereby increasing matrix stiffness and promoting fibroblast activation [130]. Enhanced LOX expression and enzymatic activity have been demonstrated in bleomycin (BLM)–induced murine models of lung fibrosis and visualized by molecular magnetic resonance imaging (MRI) using oxyamine-based probes [131]. Increased matrix stiffness and altered biomechanics are recognized as central drivers of fibrotic progres-sion, establishing a positive feedback loop between extracellular matrix remodeling and myofibroblast activation [132]. Recently, it was shown that epithelial YAP–TEAD signal-ing regulates LOX expression, and pharmacologic inhibition of this pathway in the AT2s attenuates pulmonary fibrosis in preclinical models [133].

Proprotein convertase subtilisin/kexin type 9 (PCSK9) has recently been implicated as a regulator of lipid metabolism, inflammation, and epithelial plasticity. Experimental in-hibition of PCSK9 reduces epithelial–mesenchymal transition (EMT), oxidative stress, and Wnt/β-catenin signaling in alveolar epithelial cells and attenuates fibrosis in the bleomycin (BLM) model [134-137]. PCSK9–oxidized liporoptein cholesterol (ox-LDL)-LOX has also been associated with pro-inflammatory macrophage activation and increased inflammation, oxidative stress and smooth muscle cell proliferation, and anthogenesis [135, 138]. Although direct data on LIF-specific regulation are not studied yet, these findings support a broader role for lipid pathways in coordinating epithelial differentiation, im-mune tone, and fibroblast activation. Conclusively, LOX and PCSK9 represent emerging matrix–lipid regulatory nodes that shape AT2–Fib paracrine signaling.´´

  1. While the WI model is well-framed, several sections mainly recapitulate published data without providing novel interpretation or explicit conceptual synthesis.

We would like to thank the reviewer for the comment. To address this, we now include the following paragraph.

´´Taken together, findings from the WI-38 model highlight its value not only as a human-derived mesenchymal platform but also as a conceptual bridge connecting cellular metabolism, fibroblast plasticity, and epithelial–mesenchymal interplay. Unlike isolated fibroblast or epithelial cultures, the WI-38 system enables the controlled dissection of paracrine signaling loops that influence both lipogenic and myogenic fibroblast pheno-types. Its responsiveness to pharmacologic agents pinpoints how metabolic reprogram-ming and lipid homeostasis can be leveraged to reverse fibrotic activation. This model al-so opens new perspectives for studying fibroblast dynamics across different organs [138] and could, in the future, be expanded to include additional cell types, such as immune cells, to more accurately recapitulate the complex multicellular interactions that occur under pathophysiological conditions. Therefore, beyond reproducing established find-ings, the WI-38 paradigm provides a complex, interesting, experimental framework to mechanistically link fibroblast phenotype transitions with epithelial regenerative capaci-ty, advancing a testable model for drug discovery and translational validation.´´

  1. Repetitive phrasing and extended sentences could be tightened.

We thank the reviewer for the comment. To make the address this issue we limited some sentences as follows.

´´AT2s proliferate and transiently dedifferentiate through an intermediate AT2-to-AT1 transitional state, as confirmed by lineage tracing in bleomycin-injured mice and single-cell RNA sequencing of human IPF lungs [11,62], These cells eventually differentiate into mature AT1 cells´´

´´Lung injury resolution is a gradual, multifactorial process that depends on synchronized epithelial and mesenchymal remodeling rather than a simple reversal of fibrosis.´´

Also, throughout the manuscript, we removed repetitiveness of some words to make it easier for the reader.  

  1. The abstract could be better highlighted.

We would like to thank the reviewer for the comment. The abstract is adjusted as follows

´´Single-cell RNA-sequencing has transformed our understanding of alveolar epithelial type 2 (AT2) cells and alveolar lipofibroblasts (LIFs) during lung injury and repair. Both cell types undergo dynamic transitions through intermediate states that determine whether the lung proceeds toward regeneration or fibrosis. Emerging evidence highlights reciprocal paracrine signaling between AT2/AT1 transitional cells and LIF-derived myo-fibroblasts (aMYFs) as a key regulatory axis. Among these, amphiregulin (AREG)–EGFR signaling functions as a central profibrotic pathway whose inhibition can restore alveolar differentiation and repair. The human WI-38 fibroblast model provides a practical plat-form to study the reversible LIF–MYF switch and to screen antifibrotic and pro-regenerative compounds. Candidate therapeutics including metformin, haloperidol, rosiglitazone, and FGF10 show promise in reprogramming fibroblast and epithelial states through metabolic and signaling modulation. Integrating WI-38-based assays, alveolo-sphere co-cultures, and multi-omics profiling offers a translational framework for identi-fying interventions that halt fibrosis and actively induce lung regeneration. This review highlights a unifying framework in which epithelial and mesenchymal plasticity con-verge to define repair outcomes and identifies actionable targets for promoting alveolar regeneration in chronic lung disease.´´

  1. Key knowledge gaps and translational implications should be precise

We thank the reviewer for this valuable suggestion. In the revised manuscript, we have added a dedicated paragraph as follows

´´Despite rapid advances in single-cell mapping and lineage-tracing technologies, sev-eral critical knowledge gaps remain. The precise molecular cues that govern the transition between fibroblast states, the spectrum of AT2 transitional intermediates in human lungs, and the determinants of epithelial–mesenchymal reciprocity still remain to be understood. Moreover, most mechanistic insights derive from murine injury models, highlighting the need for validation in human tissue and ex vivo platforms. From a translational stand-point, integrating in vitro models such as WI-38 fibroblast–alveolosphere co-cultures and precision-cut lung slices could accelerate the identification of context-specific antifibrotic therapies. Clarifying these regulatory checkpoints will be essential to bridge fundamental discovery with clinical intervention in chronic lung diseases such as IPF and COPD.´´

Round 2

Reviewer 2 Report

Comments and Suggestions for Authors

Congratulations to extensive revision, this contribution is of major iinsight to paracrine signaling and dynamic coordination of transitional states in the alveolar epithelial type 2 cells and  alveolar lipofibroblasts during homeostasis, injury and repair;

Author Response

Thank you very much for your kind confirmation.

Reviewer 3 Report

Comments and Suggestions for Authors

Thank you for adding the LOX and PCSK9 subsection. While the paragraph includes interesting information, it does not yet provide a clear or well-supported explanation of their roles in lipofibroblast biology or paracrine signaling. At this stage, the section is not well integrated with the rest of the manuscript and feels somewhat disconnected. If the authors cannot identify appropriate references or provide a stronger mechanistic link to support this content,  recommend removing this subsection from the revision.

Author Response

Thank you for adding the LOX and PCSK9 subsection. While the paragraph includes interesting information, it does not yet provide a clear or well-supported explanation of their roles in lipofibroblast biology or paracrine signaling. At this stage, the section is not well integrated with the rest of the manuscript and feels somewhat disconnected. If the authors cannot identify appropriate references or provide a stronger mechanistic link to support this content,  recommend removing this subsection from the revision.

We thank the reviewer for this helpful comment. To address the concern, we have replaced the previous LOX/PCSK9 subsection with a concise and accurately framed paragraph that clearly states the current limitations of the literature and avoids unsupported mechanistic claims.

´´1.12. LOX and PCSK9 as upstream modulators of the fibroblast–epithelium axis

Although no studies have directly examined LOX or PCSK9 in the context of LIF biology or AT2–LIF paracrine signaling, both pathways influence upstream processes that shape the fibrotic niche. Lysyl oxidase (LOX) increases extracellular matrix crosslinking and stiffness, which is known to enhance generic fibroblast activation and promote profibrotic mechanical feedback [129–132]. Proprotein convertase subtilisin/kexin type 9 (PCSK9), through its regulation of lipid homeostasis, oxidized LDL uptake, inflammation, and epithelial stress responses, has been shown to modulate EMT, oxidative injury, and Wnt/β-catenin signaling in alveolar epithelial cells [133–137]. While their direct involve-ment in LIF-to-MYF transitions remains untested, these pathways represent broader matrix–lipid regulatory axes that may indirectly influence epithelial–mesenchymal reciprocity in fibrotic lung remodeling. Their integration into future mechanistic studies could clarify whether they contribute to LIF or transitional AT2 cell dysfunction.´´